# Path Choice Matters for Clear Attribution in Path Methods

**Borui Zhang , Wenzhao Zheng , Jie Zhou , Jiwen Lu**[*]
Department of Automation, Tsinghua University, China
{zhang-br21, zhengwz18}@mails.tsinghua.edu.cn; {jzhou, lujiwen}@tsinghua.edu.cn

## Abstract

Rigorousness and clarity are both essential for interpretations of DNNs to engender human trust. Path methods are commonly employed to generate rigorous attributions that satisfy three axioms. However, the meaning of attributions remains ambiguous due to distinct path choices. To address the ambiguity, we introduce **Concentration Principle**, which centrally allocates high attributions to indispensable features, thereby endowing aesthetic and sparsity. We then present **SAMP**, a model-agnostic interpreter, which efficiently searches the near-optimal path from a pre-defined set of manipulation paths. Moreover, we propose the infinitesimal constraint (IC) and momentum strategy (MS) to improve the rigorousness and optimality. Visualizations show that SAMP can precisely reveal DNNs by pinpointing salient image pixels. We also perform quantitative experiments and observe that our method significantly outperforms the counterparts. [1]

## 1 Introduction

The lack of transparency in deep neural networks (DNNs) hinders our understanding of how these complex models make decisions (Bodria et al., 2021; Zhang & Zhu, 2018; Gilpin et al., 2018), which poses significant risks in safety-critical applications like autonomous driving and healthcare. Numerous interpretation methods (Zeiler & Fergus, 2014; Bach et al., 2015; Zhou et al., 2016; Selvaraju et al., 2017) have been proposed to shed light on the underlying behavior of DNNs. These methods attribute model outputs to specific input features to reveal the contributions. In this way, attribution methods serve as valuable debugging tools for identifying model or data mistakes. However, despite these efforts, users often lack confidence in attributions, which can be blamed on lack of rigorousness and clarity in current methods. Attributions are influenced by three types of artifacts (Sundararajan et al., 2017), namely data artifacts, model mistakes, and interpretation faults. To enhance user trust, it is crucial to sever the impact of the last factor.

One way to enhance the reliability of interpretations is ensuring their theoretical rigorousness. Given a complex mapping function $f : \mathcal{X} \mapsto \mathbb{R}$, we define the target point $\boldsymbol{x}^T \in \mathcal{X}$ and the baseline point $\boldsymbol{x}^0$. Interpretations aim at explaining how the baseline output $y^0$ gradually becomes $y^T$ when baseline $\boldsymbol{x}^0$ changes to $\boldsymbol{x}^T$. Early interpretation methods (Selvaraju et al., 2017; Montavon et al., 2018) employ Taylor expansion on the baseline as $y^T = y^0 + \nabla f(\boldsymbol{x}^0)^T (\boldsymbol{x}^T - \boldsymbol{x}^0) + R_1(\boldsymbol{x}^T)$. However, the local linear approximation can hardly interpret nonlinear models due to non-negligible errors of the Lagrangian remainder $R_1(\boldsymbol{x}^T)$, which makes attributions less convincing. An intuitive solution is to split the path from $\boldsymbol{x}^0$ to $\boldsymbol{x}^T$ into small segments, each of which tends to be infinitesimal. In this formulation, the variation $\Delta y$ can be formulated in integral form as $\Delta y = y^T - y^0 = \int_l \nabla f(x)^T \, \mathrm{d}\boldsymbol{x}$. The attributions $a_i$ of each feature $x_i$ is gradually accumulated through the line integral, which is commonly referred to as **path methods** (Friedman, 2004; Sundararajan et al., 2017; Xu et al., 2020; Kapishnikov et al., 2021). Game theory research (Friedman, 2004) has proved that path methods are the only method satisfying three axioms, namely dummy, additivity, and efficiency.

Ensuring rigorousness alone is insufficient for convincing interpretations. Distinct path choices in existing path methods highly impact attributions and lead to ambiguity in interpretations. Integrated Gradients (IG) (Sundararajan et al., 2017) adopts a simple straight line from $\boldsymbol{x}^0$ to $\boldsymbol{x}^T$ for symmetry. BlurIG (Xu et al., 2020) defines a path by progressively blurring the data $\boldsymbol{x}^T$ adhering to additional

---

[*]Corresponding author.
[1]Code: https://github.com/zbr17/SAMP

scale-space axioms. GuidedIG (Kapishnikov et al., 2021) slightly modifies the straight line to bypass points with sharp gradients. However, the question of which path choice is better remains unanswered. The lack of research on the optimal path selection hampers the clarity of attributions.

To the best of our knowledge, we are the first to consider the optimal path for clarity. To start with, we define the **Concentration Principle**. This principle guides the interpreter to identify the most essential features and allocate significant attributions to them, resulting in aesthetic and sparse interpretations. Subsequently, we propose **SAMP** (Salient Manipulation Path), which greedily searches the near-optimal path from a pre-defined set of manipulation paths. Moreover, we constrain the $l1$-norm of each manipulation step below an upper bound to ensure the infinitesimal condition for the line integral and employ the momentum strategy to avoid converging to local solutions. Visualizations on MNIST (Deng, 2012), CIFAR-10 (Krizhevsky et al., 2009), and ImageNet (Deng et al., 2009) demonstrate the superiority of SAMP in discovering salient pixels. We also conduct quantitative experiments and observe a clear improvement compared with other interpretation methods as shown in Figure 1. We highlight our contributions as follows:

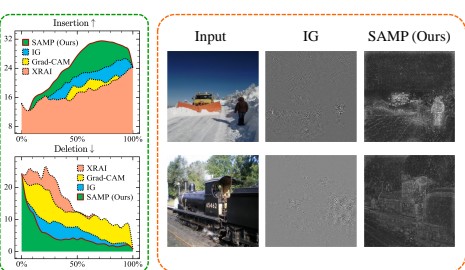

Figure 1: we propose SAMP to eliminate ambiguity of attributions by path methods, which can precisely pinpoint important pixels and produce clear saliency maps. Quantitative results show a consistent improvement in Deletion/Insertion metrics.

- **Concentration Principle for Clear Attributions.** We introduce Concentration Principle, which enhances the clarity of attributions by prioritizing sparse salient features.

- **A Model-agnostic Interpreter, SAMP.** The proposed interpreter SAMP is able to efficiently discovers the near-optimal path from a pre-defined set of manipulation paths.

- **Two Play-and-plug Auxiliary Modules.** We design infinitesimal constraint (IC) and the momentum strategy (MS) to ensure rigorousness and optimality.

- **Consistent Improvement in Explainability.** Qualitative and quantitative experiments show SAMP pinpoints salient areas accurately and consistently outperforms counterparts.

## 2 RELATED WORK

Considerable attempts expect to reveal the mysterious veil of DNNs by different techniques. Ad-hoc methods (Zhang et al., 2018b; Liang et al., 2020; Agarwal et al., 2021; Wan et al., 2020; Wang & Wang, 2021; Shen et al., 2021; Barbiero et al., 2022) try to observe or intervene in latent variables of DNNs, which rely on specific model types. On the contrary, post-hoc methods (Simonyan et al., 2014; Bach et al., 2015; Zhou et al., 2016; Selvaraju et al., 2017; Lundberg & Lee, 2017) ignore concrete implementations and focus on imitating the outside behavior. According to how attributions are generated, we mainly divide post-hoc methods into two categories: perturbation methods (Ribeiro et al., 2016; Fong & Vedaldi, 2017; Petsiuk et al., 2018) and back-propagation methods (Zeiler & Fergus, 2014; Bach et al., 2015; Selvaraju et al., 2017).

**Perturbation Methods.** An intuitive idea for attributions is to perturb the inputs and observe the output variations. Occlusion method (Zeiler & Fergus, 2014) simply covers up partial areas of input and examines the score change. LIME (Ribeiro et al., 2016) interprets the local space around the prediction by linear regression. Prediction difference analysis (Zintgraf et al., 2017) describes the output variation from a probabilistic perspective. Meaningful perturbation (Fong & Vedaldi, 2017) aims at discovering the deletion regions with compact information, which is further extended by RISE (Petsiuk et al., 2018) by the weighted average of multiple random masks. DANCE (Lu et al., 2021) introduces a subtle perturbation to input without influence on internal variables. Most perturbation methods require multiple iterations, which leads to a heavy computation burden. Moreover, most of these methods lack rigorous axiomatic guarantees.

**Back-propagation Methods.** Another kind of interpretations recovers signals or generates attributions by back-propagating information layer by layer. Early research (e.g, Deconvolution (Zeiler & Fergus, 2014) and Guided-BP (Springenberg et al., 2015)) reverses the forward procedure and

recover active signals in input space. Recent attempts generate attributions by propagating gradients (Shrikumar et al., 2016), relevance (Bach et al., 2015), and difference-from-reference (Shrikumar et al., 2017). Most methods choose gradients as the propagation intermediary for ease. Grad-CAM (Selvaraju et al., 2017) and its variants (Chattopadhay et al., 2018) directly interpolate gradients from the top layer to input size as the saliency map. SmoothGrad (Smilkov et al., 2017) aims at removing noise by averaging multiple gradients at neighbor points. The first-order Taylor decomposition (Montavon et al., 2018) assigns attributions by linearization with the gradient around the given root point. Since the difference between the data and the root is often not infinitesimal, expansion based on a single-point gradient results in a large error (Lagrangian remainder), which damages the rigorousness of interpretations. Path methods (Sundararajan et al., 2017; Xu et al., 2020; Kapishnikov et al., 2021) fix this issue by dividing the integral path into small segments. Game theory guarantees that path methods are the only method satisfying three fundamental axioms (see Proposition 1 in Friedman *et al.* (Friedman, 2004)). However, different path choices (e.g., straight line in space (Sundararajan et al., 2017) or frequency (Xu et al., 2020) and guided path along flat landscape (Kapishnikov et al., 2021)) indicate distinct attribution allocations, which makes the meaning of attribution ambiguous. Therefore, we introduce the Concentration Principle and discuss how to obtain a near-optimal path through the proposed SAMP method.

## 3 METHOD

In this section, we first summarize the canonical path methods (Friedman, 2004). Then we define Concentration Principle in Section 3.2. Subsequently, we propose the Salient Manipulation Path and derive an efficient algorithm under Brownian motion assumption in Section 3.3. Finally, we introduce infinitesimal constraint (IC) for rigorous line integrals and momentum strategy (MS) to escape from local sub-optimal solutions in Section 3.4.

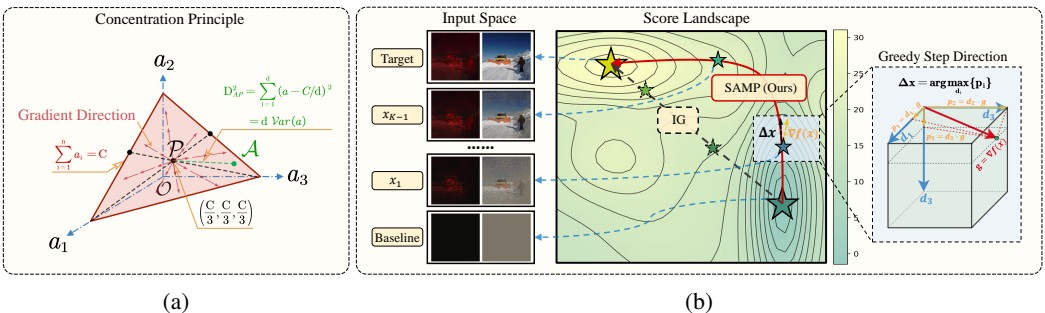

(a)                                           (b)

Figure 2: (a) Concentration Principle prioritizes attributions (green point $A$) with large distance from mean point $P$. (b) SAMP chooses the directions with max gradient projection (colored in red), and attributions allocated along this path mainly concentrate on salient pixels.

### 3.1 PRELIMINARY: PATH METHOD

Path methods (Friedman, 2004) for additive cost-sharing methods are derived from Aumann-Shapley (Aumann & Shapley, 1974), which is first introduced to machine learning by IG (Sundararajan et al., 2017). We define the many-to-one mapping as $f : \mathcal{X} \to \mathbb{R}$, where input $\boldsymbol{x}^T \in \mathcal{X}$ has $d$ features and $y^T$ denotes its output. An intuitive idea of interpreting models is to analyze how the output $y^0$ turns to $y^T$ when gradually changing baseline $\boldsymbol{x}^0$ to $\boldsymbol{x}^T$. Considering the difference between $\boldsymbol{x}^0$ and $\boldsymbol{x}^T$ is not infinitesimal, the naive Taylor decomposition $y^T = y^0 + \nabla f(\boldsymbol{x}^0)^T(\boldsymbol{x}^T - \boldsymbol{x}^0) + R_1(\boldsymbol{x}^T)$ suffers from large Lagrangian remainder $R_1(\boldsymbol{x}^T)$. Therefore, it is a natural improvement to divide path from $\boldsymbol{x}^0$ to $\boldsymbol{x}^T$ into multiple segments, which should be small enough. Assuming the model $f$ is differentiable, the output variation $\Delta y$ can be expanded as

$$\Delta y = y^T - y^0 = \int_{\rho=0}^{1} \frac{\partial f(\boldsymbol{\gamma}(\rho))}{\partial \boldsymbol{\gamma}(\rho)} \frac{\partial \boldsymbol{\gamma}(\rho)}{\partial \rho} \, \mathrm{d}\rho, \tag{1}$$

where $\boldsymbol{\gamma}(\rho)$ is path function $\boldsymbol{x} = \boldsymbol{\gamma}(\rho)$ and $\boldsymbol{\gamma}(0) = \boldsymbol{x}^0, \boldsymbol{\gamma}(1) = \boldsymbol{x}^T$. We define each feature's attribution as $a_i$ and $\Delta y$ equals sum of $a_i$ (namely completeness (Sundararajan et al., 2017)):

$$a_i \triangleq \int_{\rho=0}^{1} \frac{\partial f(\boldsymbol{\gamma}(\rho))}{\partial \gamma_i(\rho)} \frac{\partial \gamma_i(\rho)}{\partial \rho} \, \mathrm{d}\rho, \quad \Delta y = \sum_{i=1}^{d} a_i. \tag{2}$$

Game theory research (Friedman, 2004) has proved that the path method is the only interpretation method satisfying three fundamental axioms (i.e., completeness, additivity, and dummy). However, choices of path function $\gamma(\rho)$ highly impact the attribution allocation, which hampers the clarity of the interpretations. In this paper, we explore an explicit selection criterion among candidate paths.

## 3.2 CRITERION AND CANDIDATE SET

Existing path methods lack clarity due to various path choices. In Eq. (2), the attribution $\boldsymbol{a}$ is a function of the selected path $\boldsymbol{\gamma}$ as $\boldsymbol{a} = g(\boldsymbol{\gamma})$ given $\boldsymbol{x}^T, \boldsymbol{x}^0, f$. However, conventional interpretations often scatter the attributions over all pixels (Kapishnikov et al., 2021; Smilkov et al., 2017) due to unpredictable distractors. To address this, we propose the **Concentration Principle**, which introduces a selection preference for the allocation of attributions. Instead of scattering attributions across all features, we aim to concentrate them centrally on the indispensable features.

**Definition 1** (**Concentration Principle**). *A path function $\boldsymbol{\gamma}^*$ is said to satisfy Concentration Principle if the attribution $\boldsymbol{a}$ achieves the max $Var(\boldsymbol{a}) = \frac{1}{d}\sum_{i=1}^{d}(a_i - \bar{a})^2$.*

**Remark.** *Considering $\sum_{i=1}^{d} a_i$ is a constant $C = \Delta y$, the variance of $\boldsymbol{a}$ could depict the concentration degree. For a 3-feature case, this principle prefers $\boldsymbol{a} = (0.7, 0.2, 0.1)$ to $(0.4, 0.3, 0.3)$. For image input in Figure 1, this principle achieves aesthetic and sparsity. Our method clearly pinpoints important pixels while IG (Sundararajan et al., 2017) spreads attributions over most pixels. We also conduct a counting model example in Appendix A.2 to illustrate the potential challenge.*

Under this principle, we explore to formulate a tractable optimization problem. To maintain consistency in our formulation, we introduce the start point $\boldsymbol{x}^S$ and the end point $\boldsymbol{x}^E$. To approximate the line integral in Eq. (1), we use Riemann sum by dividing the path into $n$ segments as:

$$\Delta y = \sum_{k=1}^{n} \nabla f(\boldsymbol{x}^k)^T \, \mathrm{d}\boldsymbol{x}^k, \tag{3}$$

where $\mathrm{d}\boldsymbol{x}^k$ is the $k^{th}$ segment along the path and $\boldsymbol{x}^k = \boldsymbol{x}^S + \sum_{l=1}^{k} \mathrm{d}\boldsymbol{x}^l$. Analogous to Eq. (2), we calculate each attribution $a_i$ as $a_i = \sum_{k=1}^{n} (\nabla f(\boldsymbol{x}^k))_i (\mathrm{d}\boldsymbol{x}^k)_i$. However, it is intractable to directly find the optimal path from the infinite set $\Gamma$ of all path functions. Thus we construct a finite **Manipulation Path** set $\Gamma_s \subseteq \Gamma$, along which we manipulate images by inserting or deleting $s = d/n$ pixels per step. The formal definition is as follows:

**Definition 2** (**Manipulation Path**). *The $k^{th}$ segment $\mathrm{d}\boldsymbol{x}^k$ of a manipulation path $\boldsymbol{\gamma} \in \Gamma_s$ satisfies*

$$(\mathrm{d}\boldsymbol{x}^k)_i = \begin{cases} x_i^E - x_i^k, & i \in \Omega_k \\ 0, & Otherwise \end{cases}, \tag{4}$$

*where $|\Omega_k| = s$ and all $\Omega_k$ consist of a non-overlapping partition of all pixel indices which satisfy that $\forall k \neq l$, $\Omega_k \bigcap \Omega_l = \varnothing$ and $\bigcup_{k=1}^{n} \Omega_k = \{1, \cdots, d\}$.*

**Remark.** *$\Gamma_s$ is a finite set and $|\Gamma_s|$ equals to $d!/(s!)^n$.*

Following Definitions 1 and 2, we formulate the optimal path selection problem as follows:

$$\boldsymbol{\gamma}^* = \arg\max_{\boldsymbol{\gamma} \in \Gamma_s} Var(\boldsymbol{a}) = \frac{1}{d}\sum_{i=1}^{d} \left(a_i - \frac{C}{d}\right)^2. \tag{5}$$

Solving Equation 5 directly is computationally challenging. To overcome this, we propose SAMP algorithm, which leverages Brownian motion assumption to efficiently search for the optimal path.

## 3.3 SALIENT MANIPULATION PATH

The intuition of Eq. (5) is to enlarge the distance $D_{ap}$ between $\boldsymbol{a}$ (limited in the hyperplane $\sum_{i=1}^{d} a_i = C$) and the center point $(C/d, \cdots, C/d)$ in Figure 2a. We need to introduce prior knowledge to accelerate the search process. Without loss of generality, we set $s = 1$ for ease of derivation. Since the attributions are assigned sequentially, we regard the allocation process as a stochastic process [2]. We define the partial sum $u_k$ as $\sum_{i=1}^{k} a_i$ and make the following assumption:

---

[2]We use lowercase letters to denote random variables for consistency.

**Assumption 1 (Allocation as Brownian motion).** *We assume the additive process $\{u_t, t \geq 0\}$ as the Brownian motion and $u_t \sim \mathcal{N}(0, \sigma t)$ if without any constraint condition.*

We now explain the rationality of the assumption. In the model-agnostic case, we consider the input space to be isotropic. If without any constraint condition, we assume $\mathbb{E}(a_i) = \mathbb{E}(\mathrm{d}y^i) = 0$ with randomly sampled step $\mathrm{d}\boldsymbol{x}^i$ and $a_i, a_j$ for any $i \neq j$ are independent. It is important to note that we do **NOT** directly assume the $a_i, a_j$ as conditional independent given the condition $\sum_{i=1}^{d} a_i = C$. Then we assume that every $a_i$ complies with a Gaussian distribution (i.e., $a_i \sim \mathcal{N}(0, \sigma)$). If we subdivide time infinitely, the additive process $\{u_t, t \geq 0\}$ tends to a Brownian motion.

**Proposition 1.** *By Brownian motion assumption, the conditional joint distribution $P(\tilde{\boldsymbol{a}}|C) = P(a_1, \cdots, a_{d-1}|u_d = C)$ is a multivariate Gaussian distribution as:*

$$P(\tilde{\boldsymbol{a}}|C) = \frac{1}{(2\pi)^{\frac{d-1}{2}} \sqrt{|\Sigma|}} \exp\left\{ -\frac{1}{2} \left\| \tilde{\boldsymbol{a}} - \frac{C}{d}\boldsymbol{1} \right\|_{\Sigma^{-1}}^2 \right\}, \tag{6}$$

*where $\Sigma = \sigma(\boldsymbol{I} - \frac{\boldsymbol{J}}{d}) \in \mathbb{R}^{(d-1)\times(d-1)}$ and $\boldsymbol{J}$ is all-one matrix.*

**Remark.** *See proof in Appendix A.1. Eq. (6) reveals that the conditional distribution is centered at point $\mathcal{P}$ in Figure 2a. For any $i \neq j$, $Cov(a_i, a_j|u_d = C) = -\sigma/d$ indicates that allocating more to $a_i$ results in less to $a_j$. Moreover, $\mathbb{E}(u_k|u_d = C) = kC/d$ reveals that a randomly selected path tends to produce a linear variation in output. Surprisingly, we observe the curve shapes of IG (Sundararajan et al., 2017), XRAI (Kapishnikov et al., 2019), and Grad-CAM (Selvaraju et al., 2017) in Figure 1 are nearly straight lines, which is consistent with theoretical analysis.*

As the dimension of images is always high, we investigate the asymptotic property of $P(\tilde{\boldsymbol{a}}|C)$ as:

**Proposition 2.** *Since $\lim_{d\to\infty} \Sigma = \sigma\boldsymbol{I}$, conditional covariance $Cov(a_i, a_j|u_d = C)$ is nearly zero with high dimension $d$. Thus we can approximate Eq. (6) as:*

$$\hat{P}(\tilde{\boldsymbol{a}}|C) = \frac{\exp\left(-\frac{D_{ap}^2}{2\sigma}\right)}{(2\pi)^{\frac{d-1}{2}} \sqrt{|\Sigma|}} \tag{7}$$

**Remark.** *$\hat{P}(\tilde{\boldsymbol{a}}|C) = P(\tilde{\boldsymbol{a}}|C)e^{a_d^2/(2\sigma)}$. As the last attribution $a_d$ tends to 0 if $d$ is high enough, the approximation error of $\hat{P}(\tilde{\boldsymbol{a}}|C)$ is tolerable.*

Since image dimension is always high, we regard any two attributions $a_i, a_j$ as nearly independent by Proposition 2. Therefore, we can maximize each attribution separately with negligible error while reducing the computational complexity from facto-

---

**Algorithm 1:** The SAMP++ algorithm.

**Input:** Start point $\boldsymbol{x}^S$; End point $\boldsymbol{x}^E$; Upper bound $\eta$; Momentum coefficient $\lambda$.
**Output:** Attribution $\boldsymbol{a}$; Path segments $\mathbb{D}$.

1 Reset $k = 0$ and set of path segments $\mathbb{D} = \emptyset$;
2 Initialize $\boldsymbol{x}^k = \boldsymbol{x}^S$, $\boldsymbol{a}^k = \boldsymbol{0}$, $\boldsymbol{g}^k = \nabla f(\boldsymbol{x}^S)$;
3 **while** $\boldsymbol{x}^k \neq \boldsymbol{x}^E$ **do**
4     Increase index $k$ by 1;
5     Update $\boldsymbol{g}^k = \lambda\boldsymbol{g}^{k-1} + (1-\lambda)\nabla f(\boldsymbol{x}^k)$;
6     Compute $\alpha_j = g_j^k(x_j^E - x_j^k)$ if $x_j^E \neq x_j^k$ and $-\infty$ otherwise;
7     Construct $\mathbb{M}_k = \{i \mid i \in top_s\{\alpha_j\}\}$;
8     Compute $(\mathrm{d}\boldsymbol{x}^k)_i = x_i^E - x_i^k$ if $i \in \mathbb{M}_k$ and 0 otherwise;
9     If $\|\mathrm{d}\boldsymbol{x}^k\|_1 > \eta$: $\mathrm{d}\boldsymbol{x}^k = \frac{\eta}{\|\mathrm{d}\boldsymbol{x}^k\|_1}\mathrm{d}\boldsymbol{x}^k$;
10     Move current point $\boldsymbol{x}^k = \boldsymbol{x}^{k-1} + \mathrm{d}\boldsymbol{x}^k$;
11     Update attribution $\boldsymbol{a}^k = \boldsymbol{a}^{k-1} + \boldsymbol{g}^k \cdot \mathrm{d}\boldsymbol{x}^k$;
12     Expand $\mathbb{D} = \mathbb{D}\bigcup\{\mathrm{d}\boldsymbol{x}^k\}$;
13 Return $\boldsymbol{a}^k, \mathbb{D}$.

---

rial $\mathcal{O}(d!)$ to linear $\mathcal{O}(d)$. Specifically, we choose $s$ pixels with the largest projection of gradient $\nabla f(\boldsymbol{x}^k)$ onto $\mathrm{d}\boldsymbol{x}^k$. We name this greedy selection strategy as **Salient Manipulation Path (SAMP)** and take insertion direction as an example to formulate SAMP as:

$$(\mathrm{d}\boldsymbol{x}^k)_i = \begin{cases} x_i^E - x_i^k, & i \in \mathbb{M}_k \\ 0, & \text{Otherwise} \end{cases}, \tag{8}$$

where $\mathbb{M}_k = \{i \mid i \in top_s\{\alpha_j\}\}$ ($top_s(\cdot)$ means the largest $s$ elements) and $\alpha_j = (\nabla f(\boldsymbol{x}^k))_j(x_j^E - x_j^k)$ if $x_j^E \neq x_j^k$ and $-\infty$ otherwise. It is obvious that the path defined above belongs to $\Gamma_s$.

### 3.4 TOWARDS RIGOROUSNESS AND OPTIMALITY

Two potential issues still remain in our proposed SAMP interpreter. First, if the step size $|\mathrm{d}\boldsymbol{x}^k|$ is too large, the infinitesimal condition may be violated, thereby breaking the completeness axiom in

Eq. (2). Besides, most greedy algorithms tend to get stuck in the local sub-optimal solution. To address these, we propose the infinitesimal constraint and the momentum strategy respectively.

**Infinitesimal Constraint (IC).** To ensure the completeness axiom, we need to restrict each step size below a given bound $\eta > 0$. Therefore we rectify $\mathrm{d}\boldsymbol{x}^k$ in Eq. (8) as:

$$\mathrm{d}\hat{\boldsymbol{x}}^k = \begin{cases} \dfrac{\eta}{\|\mathrm{d}\boldsymbol{x}^k\|_1}\,\mathrm{d}\boldsymbol{x}^k, & \text{if } \|\mathrm{d}\boldsymbol{x}^k\|_1 > \eta \\ \mathrm{d}\boldsymbol{x}^k, & \text{Otherwise} \end{cases} \tag{9}$$

Note that the above constraint does not affect the convergence of SAMP. According to the definition of manipulation paths, it is easy to know the sum of L1 norm of all steps is a constant value as $\sum_{k=1}^{n}\|\mathrm{d}\boldsymbol{x}^k\|_1 = \|\boldsymbol{x}^S - \boldsymbol{x}^E\|_1 = C$. As long as $\eta > 0$, the constrained SAMP can certainly converge after finite iterations.

**Momentum Strategy (MS).** Due to the nature of greedy algorithms, SAMP runs the risk of falling into a local optimum. Inspired by the gradient descent with momentum, we incorporate the momentum strategy to coast across the flat landscape through the inertia mechanism as follows:

$$\boldsymbol{g}^k = \lambda \boldsymbol{g}^{k-1} + (1-\lambda)\nabla f(\boldsymbol{x}^k). \tag{10}$$

By substituting $\mathrm{d}\boldsymbol{x}^k$ with $\mathrm{d}\hat{\boldsymbol{x}}^k$ and $\nabla f(\boldsymbol{x}^k)$ with $\boldsymbol{g}^k$, we formulate SAMP++ in Algorithm 1.

# 4 EXPERIMENT

In this section, we conduct qualitative and quantitative experiments to demonstrate the superiority of our proposed SAMP method. Due to the wide variety of interpretability methods, they often need to be evaluated from multiple dimensions (Nauta et al., 2022). Our proposed SAMP method belongs to attribution methods. We first perform qualitative experiments to verify the Concentration Principle claimed above and compare the visualization results with other counterparts in Section 4.2. Subsequently, we employ Deletion/Insertion metrics (Petsiuk et al., 2018) to examine SAMP quantitatively and conduct a completeness check with the Sensitivity-N metric (Ancona et al., 2018) in Section 4.3. Extensive ablation studies demonstrate the effectiveness of each feature in Section 4.4.

## 4.1 EXPERIMENTAL SETTING

**Datasets and Models.** We evaluate SAMP on the widely used MNIST (Deng, 2012), CIFAR-10 (Krizhevsky et al., 2009), and ImageNet (Deng et al., 2009). For MNIST and CIFAR-10 datasets, we simply build two five-layer CNNs (c.f. Appendix A.3) and train them to convergence using AdamW optimizer (Loshchilov & Hutter, 2017). For ImageNet dataset, we use the pre-trained ResNet-50 model (He et al., 2016) from PyTorch torchvision package (Paszke et al., 2019).

**Metrics.** Interpretations should faithfully reveal the attention of model decisions. One evaluation for judging attributions is to check whether features with large attribution have a significant effect outputs. Therefore, we choose the Deletion/Insertion metrics (Petsiuk et al., 2018) for quantitative comparison. We delete/insert pixels sequentially in the descending order of attributions, plot the output curve, and calculate the area under the curve (AUC). For Deletion, a smaller AUC indicates better interpretability; for insertion, a larger AUC is expected. Moreover, we wish to examine the effect of the infinitesimal constraint (IC) on the rigorousness of SAMP. Therefore, we adopt the Sensitivity-N metric (Ancona et al., 2018) by calculating the Pearson correlation between the sum of attributions and the model output for completeness check.

**Implementation Details.** We compare Deletion/Insertion metrics of SAMP with 12 mainstream interpretation methods.[3] Following the configuration (Petsiuk et al., 2018), we set the baseline point as a zero-filled image for Deletion and a Gaussian-blurred image for Insertion. We randomly select 100 images from each dataset and report the mean and standard deviation of AUCs. Specifically, for MNIST and CIFAR-10, we set the Gaussian blur kernel size $s_g$ to 11 the variance $\sigma_g$ to 5, and the step size for calculating metrics $s_m$ to 10; for ImageNet, $s_g = 31$, $\sigma_g = 5$, and $s_m = 224 \times 8$. If without special specifications, we fix the step size $s$ in SAMP as $224 \times 16$ for ImageNet and 10 for other datasets, the ratio of the infinitesimal upper bound $\eta$ to $\|\Delta\boldsymbol{x}\|_1$ as 0.1, and the momentum coefficient $\lambda$ as 0.5. We perform all experiments with PyTorch on one NVIDIA 3090 card.

---

[3]The benchmark code will be released together with SAMP.

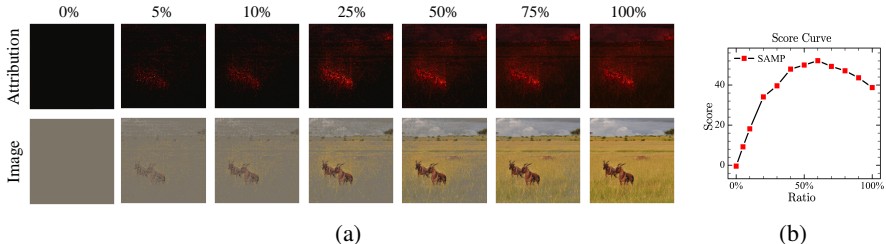

(a)                                                                (b)

Figure 3: Verification of Concentration Principle. (a) Visualizations of intermediate points and corresponding attributions along the path solved by SAMP. (b) The output score curve from the baseline point to the target image.

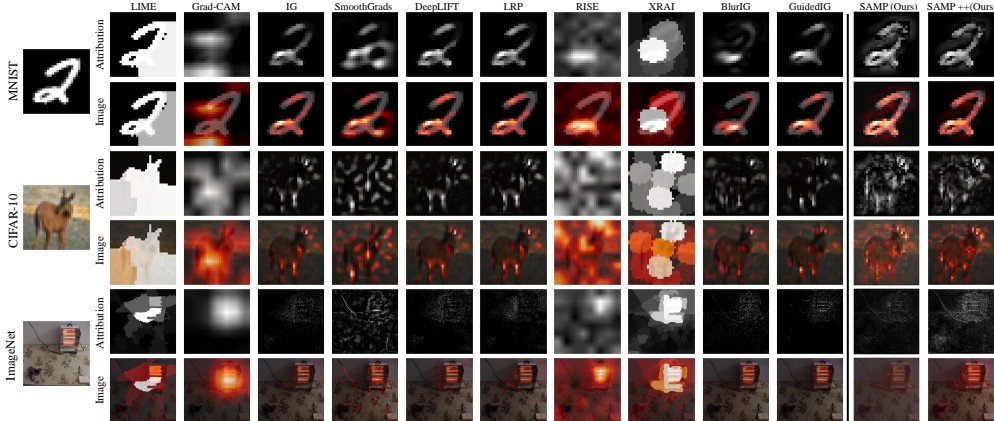

Figure 4: Visualizations on MNIST, CIFAR-10, and ImageNet compared with other methods.

## 4.2 QUALITATIVE VISUALIZATION

### 4.2.1 VERIFICATION OF PROPERTIES

We first verify whether SAMP can reach an expected path towards Concentration Principle. For clear visualization, we set the baseline point as zero-filled, and choose the manipulation direction from $x^0$ to $x^T$. Along the path solved by SAMP, the intermediate points and corresponding attributions at different stages are visualized separately, as shown in Figure 3a. We can see that the first 25% of the path has precisely pinpointed the subject animal. Besides, we plot the output scores at different stages along the manipulation path in Figure 3b. A rapid rise can be observed at the start, which indicates that SAMP tends to capture the most salient pixels first. At the same time, there is a small drop at the end. We ascribe this to background pixels, which interfere with the output score.

### 4.2.2 VISUALIZATION COMPARISON

We compare the visualization results of SAMP with other mainstream interpretation methods (Ribeiro et al., 2016; Selvaraju et al., 2017; Sundararajan et al., 2017; Smilkov et al., 2017; Shrikumar et al., 2017; Bach et al., 2015; Petsiuk et al., 2018; Kapishnikov et al., 2019; Xu et al., 2020; Kapishnikov et al., 2021). After randomly selecting input images on MNIST, CIFAR-10, and ImageNet, we calculate the attribution results of each method. We first convert the attributions to a grayscale image for visualization and also superimpose the attribution values with the original image. Figure 4 shows the comparison of the SAMP method with existing methods. As can be seen, the attribution results allocated by our method pinpoint important pixels and localize all pixels on salient objects most completely. Additionally, the attribution results of the SAMP++ approach are broadly similar to SAMP, but the results of SAMP++ are more fine-grained due to the infinitesimal constraints (for instance, the subject is more separated from the background).

## 4.3 QUANTITATIVE ANALYSIS

We conducted quantitative experiments to assess the performance of SAMP, including metrics such as Deletion/Insertion and Sensitivity-N check. In addition, we also carried out evaluations such as $\mu$Fidelity (Novello et al., 2022) and pointing game (Zhang et al., 2018a) in Section A.5.4.

Table 1: Deletion/Insertion metrics on MNIST, CIFAR-10, and ImageNet.

| Method | MNIST | | CIFAR-10 | | ImageNet | |
|---|---|---|---|---|---|---|
| | Deletion↓ | Insertion↑ | Deletion↓ | Insertion↑ | Deletion↓ | Insertion↑ |
| LRP | -0.003 (±0.13) | **0.808** (±**0.10**) | -0.257 (±0.49) | **1.452** (±**0.37**) | 0.210 (±0.13) | 0.575 (±0.15) |
| CAM | 0.221 (±0.15) | 0.715 (±0.11) | 0.314 (±0.31) | 0.863 (±0.23) | 0.313 (±0.129) | 0.897 (±0.13) |
| LIME | 0.282 (±0.14) | 0.597 (±0.09) | 0.479 (±0.29) | 0.722 (±0.24) | 0.312 (±0.13) | **0.898** (±**0.14**) |
| Grad-CAM | 0.221 (±0.15) | 0.715 (±0.11) | 0.314 (±0.31) | 0.863 (±0.23) | 0.313 (±0.13) | 0.897 (±0.13) |
| IG | -0.038 (±0.14) | 0.795 (±0.11) | **-0.372** (±**0.54**) | **1.452** (±**0.40**) | 0.197 (±0.13) | 0.725 (±0.20) |
| SmoothGrads | 0.003 (±0.13) | 0.547 (±0.11) | 0.777 (±0.55) | 0.517 (±0.28) | 0.300 (±0.13) | 0.605 (±0.17) |
| DeepLIFT | -0.025 (±0.14) | 0.791 (±0.11) | -0.300 (±0.51) | 1.443 (±0.38) | 0.216 (±0.12) | 0.688 (±0.18) |
| RISE | 0.059 (±0.11) | 0.651 (±0.12) | 0.149 (±0.35) | 0.904 (±0.27) | 0.282 (±0.13) | 0.849 (±0.15) |
| XRAI | 0.120 (±0.12) | 0.754 (±0.10) | 0.248 (±0.33) | 0.910 (±0.21) | 0.346 (±0.16) | 0.865 (±0.14) |
| Blur IG | 0.021 (±0.02) | 0.804 (±0.17) | -0.107 (±0.39) | 1.407 (±0.47) | 0.261 (±0.14) | 0.712 (±0.22) |
| Guided IG | **-0.041** (±**0.14**) | 0.762 (±0.10) | -0.276 (±0.47) | 1.209 (±0.35) | **0.167** (±**0.13**) | 0.699 (±0.21) |
| SAMP (ours) | **-0.093** (±**0.14**) | **1.074** (±**0.18**) | **-0.733** (±**0.67**) | **1.458** (±**0.40**) | **0.154** (±**0.12**) | **0.984** (±**0.20**) |
| SAMP++ (ours) | **-0.137** (±**0.151**) | **1.050** (±**0.18**) | **-0.899** (±**0.72**) | **1.514** (±**0.43**) | **0.145** (±**0.12**) | **1.116** (±**0.24**) |

Figure 5: Sensitivity-N check for IC.

Figure 6: Impact of momentum coefficient $\lambda$.

### 4.3.1 DELETION/INSERTION COMPARISON

To precisely compare the performance, we calculate the Deletion/Insertion metrics (Petsiuk et al., 2018). We randomly sampled 100 images and report the mean and standard deviation of the AUCs (see Table 1), where "SAMP" represents the original algorithm described in Eq. (8) and "SAMP++" denotes Algorithm 1 with the infinitesimal constraint (IC) and momentum strategy (MS). Our method consistently outperforms all other methods on three datasets. We ascribe this to Concentration Principle that facilitates our method to perform clear saliency rankings. In addition, the improved version significantly improves the original one in most cases. We believe that the momentum strategy plays an essential role in prompting the algorithm to break free from the local point (c.f. Section 4.4 for ablation studies.).

### 4.3.2 SENSITIVITY-N CHECK

In this part, we show the importance of the infinitesimal constraint (IC) on rigorousness (or completeness (Sundararajan et al., 2017)). Sensitivity-N (Ancona et al., 2018) checks the completeness by calculating the Pearson correlation of $\sum_j a_j$ and $\Delta y$. We gradually increase $\beta = \|x\|_1/\eta$ (i.e., decrease the upper bound $\eta$ in Eq. (9)) and draw the curve of the correlation w.r.t. $\beta$ (see Figure 5). With the decrease of $\eta$, the correlation increases significantly. This is because IC limits each step to be infinitesimal, which ensures that Lagrangian remainder tends to 0, thereby enhancing rigorousness of Eq. (3). Interestingly, Figure 5a shows that with the further decrease of $\eta$, the numerical error becomes the main error source, and the correlation no longer rises; because $\eta$ is not small enough at the start of Figure 5b, most steps are not cropped, thereby leading to a flat correlation curve.

### 4.4 ABLATION STUDY

### 4.4.1 INFLUENCE OF IC AND MS

We perform ablation studies on the infinitesimal constraint (IC) and momentum strategy (MS), as shown in Table 2. As we can see, the improvement of SAMP in Deletion/Insertion metrics mainly comes from MS. According to Figure 6, SAMP achieves the largest improvement when $\lambda \approx 0.3$. Table 3 shows that IC has no significant impact on Deletion/Insertion metrics, which can be attributed to the fact that IC is primarily designed to maintain rigor and lacks a direct connection with

Table 2: Ablation study on IC and MS.

| Setting | ImageNet | |
|---|---|---|
| | Deletion↓ | Insertion↑ |
| SAMP | 0.154 (±0.118) | 0.984 (±0.195) |
| +MS | **0.144** (±0.115) | **1.088** (±0.251) |
| +IC | 0.159 (±0.121) | 1.056 (±0.185) |
| +MS/IC | **0.145** (±0.116) | **1.116** (±0.241) |

Table 3: Influence of upper bound $\eta$

| Bound$^{-1}$ | ImageNet | |
|---|---|---|
| ($\|\Delta \boldsymbol{x}\|_1/\eta$) | Deletion↓ | Insertion↑ |
| 1/10 | **0.159** (±0.121) | 1.056 (±0.185) |
| 1/50 | 0.218 (±0.133) | **1.130** (±0.168) |
| 1/100 | 0.249 (±0.142) | 1.031 (±0.155) |
| 1/200 | 0.279 (±0.147) | 0.939 (±0.159) |

enhancing metrics. In addition, smaller $eta$ (e.g., $\eta = 1/200$) leads to finer-grained visualization (see Figure 8), which is due to the shortened step size that focuses more on details.

### 4.4.2 CHOICE OF BASELINE POINTS

We wonder whether different baseline choices affect the performance of SAMP. Therefore, we set four different sets of baseline points with $\eta = \|\boldsymbol{x}\|_1/50$. "B" means padding with zero, "W" means padding with one, "U" denotes uniformly random initialization, and "G" denotes Gaussian random initialization. In the symbol "X+Y", X represents the deletion direction, and Y represents the insertion direction. Figure 7 shows that the impact of different baselines on the explanations is not significant compared to different methods in Figure 4. Specifically, dif-

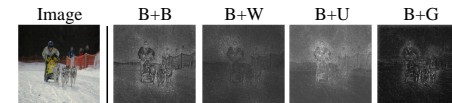

Figure 7: Visualization of different baseline.

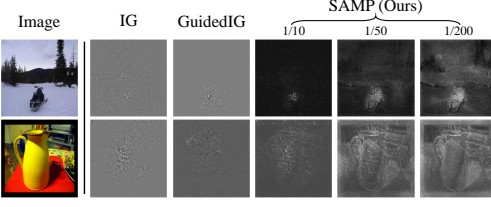

Figure 8: Results with different upper bound $\eta$.

ferent baselines have little impact on the contour information, but they do significantly affect the overall intensity (e.g., brightness), which leads to visual differences.

### 4.4.3 CHOICE OF PATHS

The choice of path also has a certain influence on Deletion/Insertion (as shown in Table 4). We discover that only using the path $\boldsymbol{x}^T \rightarrow \boldsymbol{x}^0$ achieves the best Deletion; only using $\boldsymbol{x}^0 \rightarrow \boldsymbol{x}^T$ reaches the highest Insertion. We actually use both directions at the same time, and sum attributions generated by two directions to obtain a trade-off between two metrics.

Table 4: Influence of path choices.

| Path | | ImageNet | |
|---|---|---|---|
| to $\boldsymbol{x_0}$ | to $\boldsymbol{x_T}$ | Deletion↓ | Insertion↑ |
| ✓ | | **0.108** (±0.107) | 0.659 (±0.171) |
| | ✓ | 0.199 (±0.135) | **1.330** (±0.230) |
| ✓ | ✓ | 0.159 (±0.121) | 1.056 (±0.185) |

## 5 CONCLUSION

To obtain user trust, interpretations should possess rigorousness and clarity. Even though path method (Sundararajan et al., 2017) identifies fundamental axioms for rigorousness, attributions remain ambiguous due to indeterminate path choices. In this paper, we first define **Concentration Principle**. Subsequently, we propose **Salient Manipulation Path (SAMP)**, which is a fast and greedy interpreter for solving the approximate optimal path efficiently, To enhance the rigorousness and optimality of SAMP, we propose the infinitesimal constraint (IC) and momentum strategy (MS) respectively. Visualization experiments show that the attribution generated by our method accurately discovers significant pixels and completely pinpoints all pixels of salient objects. Quantitative experiments demonstrate that our method significantly outperforms the current mainstream interpretation methods. Moreover, qualitative experiments also reveal that SAMP can obtain higher-quality semantic segmentations by visualizing the attribution values only employing class-level annotations. Investigating the utility of saliency-based explanations in annotation-limited tasks (such as weakly-supervised object recognition) and other promising domains (with a focus on NLP and medical image analysis) represents an exciting direction for further study.

ACKNOWLEDGEMENT

This work was supported in part by the National Key Research and Development Program of China under Grant 2022ZD0160102, and in part by the National Natural Science Foundation of China under Grant 62125603, Grant 62321005, and Grant 62336004.

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

## A APPENDIX

### A.1 THEORETICAL SUPPLEMENTARY

#### A.1.1 PROOF OF PROPOSITION 1

*Proof.* By assumption, $\{u_t, t \geq 0\}$ is a Brownian motion and $u_t \sim \mathcal{N}(0, \sigma t)$:

$$P(u_t) = \frac{1}{\sqrt{2\pi\sigma t}} \exp\left\{-\frac{u_t^2}{2\sigma t}\right\}$$

To treat attributions equally, we consider the subset $\{u_t, t \in \mathbb{N}\}$ with uniform time intervals ($u_0 = 0$). According to the Markov property, we can get $\forall\, k \in \mathbb{N}, u_k - u_{k-1} \sim \mathcal{N}(0, \sigma)$. Thus we can formulate the joint probability distribution as follows:

$$
\begin{aligned}
&P(u_1, u_2, \cdots, u_d) \\
=&\prod_{k=1}^{d} P(u_k | u_{k-1}, \cdots, u_1) \\
=&\prod_{k=1}^{d} P(u_k - u_{k-1} | u_{k-1}) \\
=&\prod_{k=1}^{d} \frac{1}{\sqrt{2\pi\sigma}} \exp\left\{-\frac{(u_k - u_{k-1})^2}{2\sigma}\right\} \\
=&\frac{1}{(2\pi\sigma)^{\frac{d}{2}}} \exp\left\{-\sum_{k=1}^{d} \frac{(u_k - u_{k-1})^2}{2\sigma}\right\}.
\end{aligned}
\tag{11}
$$

As $\forall k \in \{1, \cdots, d\}, a_k = u_k - u_{k-1}$, we have

$$
\begin{aligned}
(a_1, a_2, \cdots, a_d) &= \boldsymbol{J}(u_1, u_2, \cdots, u_d) \\
&= \begin{bmatrix} 1 & 0 & 0 & \cdots & 0 \\ -1 & 1 & 0 & \cdots & 0 \\ 0 & -1 & 1 & \cdots & 0 \\ \vdots & \vdots & \vdots & \ddots & \vdots \\ 0 & 0 & 0 & \cdots & 1 \end{bmatrix} \begin{bmatrix} u_1 \\ u_2 \\ u_3 \\ \vdots \\ u_d \end{bmatrix}.
\end{aligned}
$$

Thus $P(a_1, a_2, \cdots, a_d) = P(u_1, u_2, \cdots, u_d)/|\boldsymbol{J}| = P(u_1, u_2, \cdots, u_d)$. If dividing $P(a_1, a_2, \cdots, a_d)$ by $P(u_d)$, we get the conditional distribution $P(\tilde{\boldsymbol{a}}|C) = P(a_1, \cdots, a_{d-1}|u_d = C)$ as follows:

$$
\begin{aligned}
&P(a_1, a_2, \ldots, a_{d-1} | u_d = C) \\
=&P(a_1, a_2, \cdots, a_d | u_d = C) \\
=&P(a_1, a_2, \cdots, a_d)/P(u_d = C) \\
=&\frac{\sqrt{d}}{(2\pi\sigma)^{\frac{d-1}{2}}} \exp\left\{\frac{u_d^2}{2\sigma d} - \sum_{k=1}^{d} \frac{a_k^2}{2\sigma}\right\} \\
=&\frac{\sqrt{d}}{(2\pi\sigma)^{\frac{d-1}{2}}} \exp\left\{\frac{C^2}{2\sigma d} - \frac{(C - \sum_{k=1}^{d-1} a_k)^2}{2\sigma} - \sum_{k=1}^{d-1} \frac{a_k^2}{2\sigma}\right\} \\
=&\frac{\sqrt{d}}{(2\pi\sigma)^{\frac{d-1}{2}}} \exp\left\{\frac{-1}{2\sigma}\left[\tilde{\boldsymbol{a}}^T \tilde{\boldsymbol{a}} + (C - \mathbf{1}^T \tilde{\boldsymbol{a}})^2 - \frac{C^2}{d}\right]\right\} \\
=&\frac{\sqrt{d}}{(2\pi\sigma)^{\frac{d-1}{2}}} \exp\left\{\frac{-1}{2\sigma}\left[\tilde{\boldsymbol{a}}^T (\boldsymbol{I} + \mathbf{1}\mathbf{1}^T)\tilde{\boldsymbol{a}} - 2C\mathbf{1}\tilde{\boldsymbol{a}} + \frac{(d-1)C^2}{d}\right]\right\}.
\end{aligned}
\tag{12}
$$

Let $\Sigma = \frac{\sigma}{d}(d\boldsymbol{I} - \boldsymbol{1}\boldsymbol{1}^T) \in \mathbb{R}^{(d-1)\times(d-1)}$. We can easily get that $|\Sigma| = \frac{\sigma^{d-1}}{d}$ and $\Sigma^{-1} = \frac{1}{\sigma}(\boldsymbol{I} + \boldsymbol{1}\boldsymbol{1}^T)$. Thus we can simplify Eq. (12) as:

$$
\begin{aligned}
&P(a_1, a_2, \ldots, a_{d-1}|u_d = C) \\
=&\frac{\sqrt{d}}{(2\pi\sigma)^{\frac{d-1}{2}}} \exp\left\{-\frac{1}{2}\left[\tilde{\boldsymbol{a}}^T\Sigma^{-1}\tilde{\boldsymbol{a}} - 2\frac{C}{d}\boldsymbol{1}\Sigma^{-1}\tilde{\boldsymbol{a}} + \frac{C^2}{d^2}\boldsymbol{1}^T\Sigma^{-1}\boldsymbol{1}\right]\right\} \\
=&\frac{\sqrt{d}}{(2\pi\sigma)^{\frac{d-1}{2}}} \exp\left\{-\frac{1}{2}\left(\tilde{\boldsymbol{a}} - \frac{C}{d}\boldsymbol{1}\right)^T \Sigma^{-1}\left(\tilde{\boldsymbol{a}} - \frac{C}{d}\boldsymbol{1}\right)\right\} \\
=&\frac{1}{(2\pi)^{\frac{d-1}{2}}\sqrt{|\Sigma|}} \exp\left\{-\frac{1}{2}\left\|\tilde{\boldsymbol{a}} - \frac{C}{d}\boldsymbol{1}\right\|_{\Sigma^{-1}}^2\right\}.
\end{aligned}
\tag{13}
$$

Therefore, $\tilde{\boldsymbol{a}}|u_d \sim \mathcal{N}(\frac{C}{d}\boldsymbol{1}, \Sigma)$ where $\Sigma = \frac{\sigma}{d}(d\boldsymbol{I} - \boldsymbol{1}\boldsymbol{1}^T)$. $\qquad\square$

### A.1.2 SUPPLEMENTARY OF PROPOSITION 2

**(1) How to derive Eq. (7):**  We can reformulate Eq. (13):

$$
\begin{aligned}
&P(\tilde{\boldsymbol{a}}|C) = P(a_1, a_2, \ldots, a_{d-1}|u_d = C) \\
=&\frac{\sqrt{d}}{(2\pi\sigma)^{\frac{d-1}{2}}} \exp\left\{-\frac{1}{2\sigma}\left(\tilde{\boldsymbol{a}} - \frac{C}{d}\boldsymbol{1}\right)^T\left(\tilde{\boldsymbol{a}} - \frac{C}{d}\boldsymbol{1}\right)\right. \\
&\left. -\frac{1}{2\sigma}\left(\tilde{\boldsymbol{a}} - \frac{C}{d}\boldsymbol{1}\right)^T \boldsymbol{1}\boldsymbol{1}^T\left(\tilde{\boldsymbol{a}} - \frac{C}{d}\boldsymbol{1}\right)\right\} \\
=&\frac{\sqrt{d}}{(2\pi\sigma)^{\frac{d-1}{2}}} \exp\left\{-\frac{D_{ap}^2}{2\sigma} - \frac{1}{2\sigma}\left(C - \sum_{k=1}^{d-1} a_k\right)^2\right\} \\
=&\frac{\sqrt{d}}{(2\pi\sigma)^{\frac{d-1}{2}}} \exp\left\{-\frac{D_{ap}^2 + a_d^2}{2\sigma}\right\} \\
=&\hat{P}(\tilde{\boldsymbol{a}}|C)e^{-a_d^2/(2\sigma)},
\end{aligned}
\tag{14}
$$

where $\hat{P}(\tilde{\boldsymbol{a}}|C)$ is the estimation of $P(\tilde{\boldsymbol{a}}|C)$. By Proposition 1 we know that $a_d|u_d \sim \mathcal{N}(\frac{C}{d}, \sigma\frac{d-1}{d})$. Although the expectation of $a_d$ gradually approaches zero with the increase of $d$, the variance gradually approaches $\sigma$. Thus the approximation error cannot be completely zero. Fortunately, the experimental results show that the attribution allocation is not completely homogeneous. Figure 3b reveals that the last allocated $a_d$ is usually approximately zero due to the saturation region. Therefore, the approximation error in Proposition 2 is generally acceptable.

**(2) Element-wise limit:**  Note that $\lim_{d\to} \Sigma = \sigma\boldsymbol{I}$ actually means the element-by-element limit rather than the limit in terms of the matrix norm. Given $\Sigma(d) = \frac{\sigma}{d}(d\boldsymbol{I} - \boldsymbol{1}\boldsymbol{1}^T)$, we can easily get that

$$
\forall i, j \in \mathbb{N}, \lim_{d\to\infty} \Sigma_{ij} = \begin{cases} \sigma, & i = j \\ 0, & i \neq j \end{cases}.
\tag{15}
$$

It is hard to find a norm $\|\cdot\|$ such that $\lim_{d\to\infty} \|\Sigma - \sigma\boldsymbol{I}\| = 0$. For example, $\|\Sigma - \sigma\boldsymbol{I}\|_{m_\infty} = (d-1)\max_{ij}|\Sigma_{ij} - \sigma\boldsymbol{I}_{ij}|$ is always 1, not tending to 0. Therefore, Proposition 2 has certain limitations. The continuous mapping such as the determinant $(\det(\cdot))$ and the limit operation $(\lim_{d\to\infty}(\cdot))$ are not commutative.

### A.2 POTENTIAL CHALLENGE OF CONCENTRATION PRINCIPLE

As a path method, SAMP generates attributions with three axioms (linearity, dummy, and efficiency). These axioms ensure that SAMP does not blindly pursue high variance path and only ignores ambiguous elements rather than essential ones. We visualize a blue-pixel counting model in Fig.9, and SAMP achieves consistent attributions.

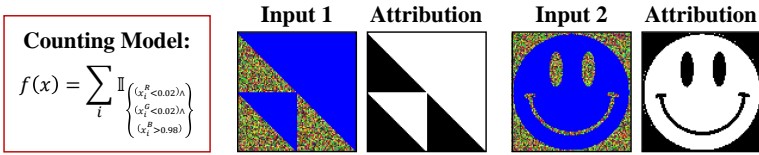

Figure 9: Counting model examples for Concentration Principle.

## A.3 MODEL ARCHITECTURE

The architecture of 5-layer CNNs adopted on the MNIST (Deng, 2012) and CIFAR-10 (Krizhevsky et al., 2009) datasets is shown in Table 5. We employ the most common ReLU activation function and an adaptive max-pooling layer to reduce the spatial dimension to 1.

Table 5: CNNs for MNIST and CIFAR-10.

| Layer | 5-layer CNNs | |
|---|---|---|
| | for MNIST | for CIFAR-10 |
| | Input $1 \times 28 \times 28$ | Input $3 \times 32 \times 32$ |
| conv1 | $5 \times 5, 8$ + ReLU | $5 \times 5$ + ReLU |
| conv2 | $2 \times 2, 24$, stride 2 | $2 \times 2, 64$, stride 2 |
| conv3 | $4 \times 4, 288$ + ReLU | $4 \times 4, 512$ + ReLU |
| conv4 | $2 \times 2, 864$, stride 2 | $2 \times 2, 1536$, stride 2 |
| conv5 | $3 \times 3, 2592$ + ReLU | $3 \times 3, 4608$ + ReLU |
| pool | adaptive max pool | |

## A.4 EVALUATION DETAIL

### A.4.1 EVALUATION PROTOCOL

We employ classification models for Deletion/Insertion evaluation. Conventional classification models output the final scores with a soft-max layer as

$$\tilde{y}_i = \frac{e^{y_i}}{\sum_j e^{y_j}}.$$

However, $\tilde{y}_i$ is coupled with the output score $y_j$ ($j \neq i$) of other classes. Therefore, we directly utilize the output score $y_i$ before the soft-max layer to compute metrics.

Given any input data $x$, we employ the interpretation method to obtain the attribution $a$. We first rank the attribution $a$ in descending order. Then we select $s_m$ pixels to delete or insert according to the sorting, so as to obtain a series of output scores as:

$$y = \begin{cases} \{y^T, y^{K-1}, \cdots, y^1, y^0\} & \text{For Deletion} \\ \{y^0, y^1, \cdots, y^{K-1}, y^T\} & \text{For Insertion} \end{cases}.$$

where $y^0$ and $y^T$ are path-independent values. We can normalize the series of scores as

$$\hat{y}^k = y^k/y^T,$$

and the normalized area under the curve is computed as $s_{auc} = \frac{1}{K+1} \sum_{i=0}^{K} \hat{y}$. Note that Note that $s_{auc}$ is not necessarily constrained between 0 and 1. For Deletion, the intermediate $y^k$ may be greater than $y^T$; for insertion, the intermediate $y^k$ may be less than 0.

### A.4.2 HYPERPARAMETER CHOICE

$\lambda$ **setting:** $\lambda \approx 0.3$ in Section 4.4.1 is only optimal on ImageNet, so we set $\lambda = 0.5$ for all three datasets in Section 4.1.

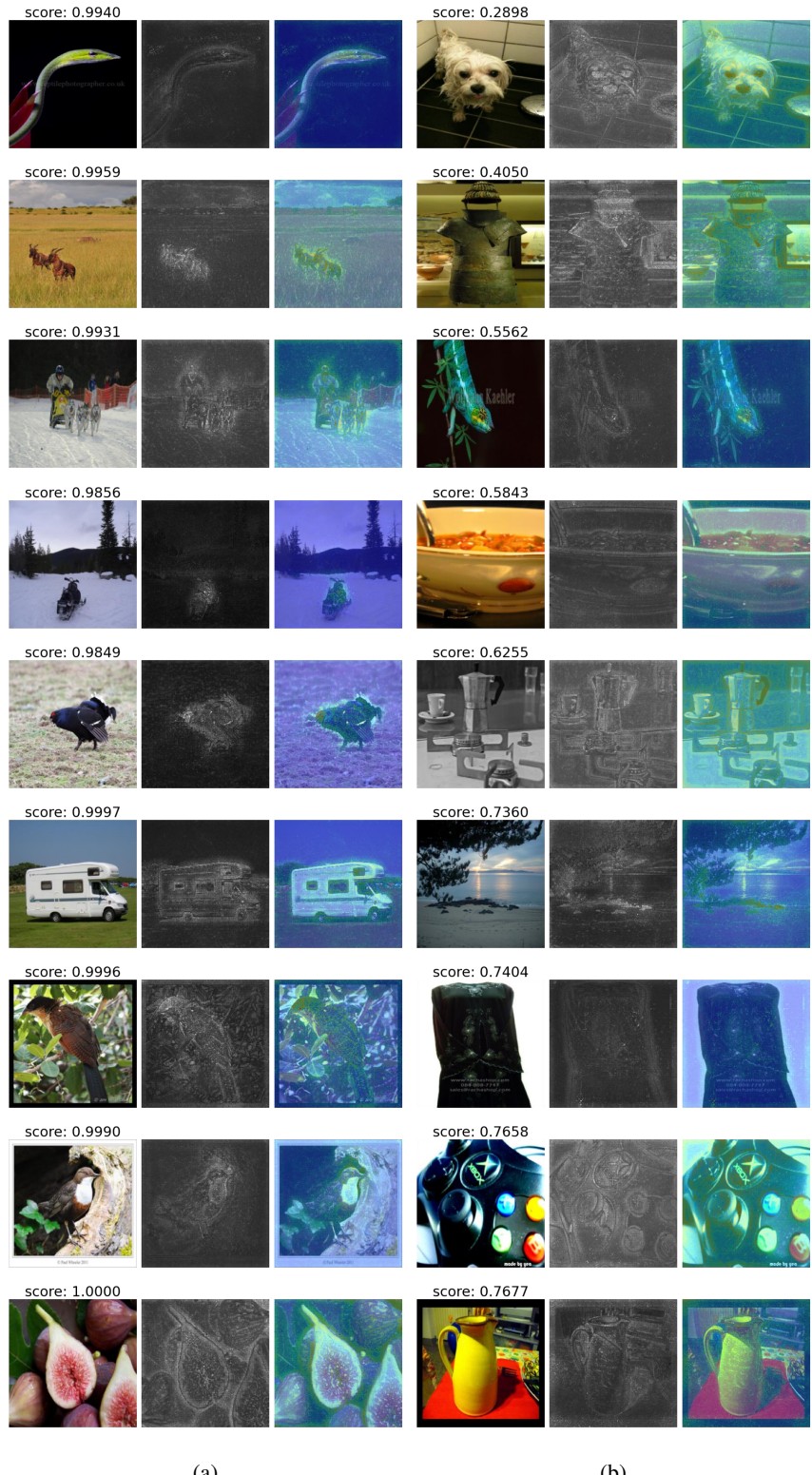

(a)             (b)

Figure 10: (a) Visualization results of images with high classification confidence. (b) Images with low classification confidence.

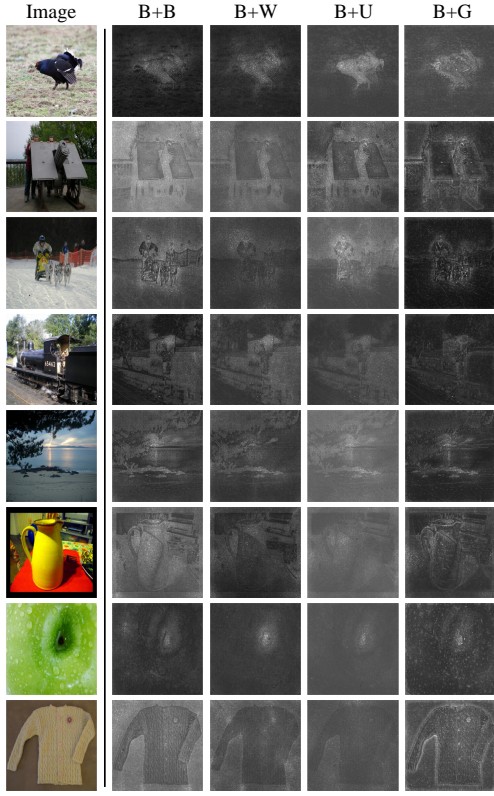

Figure 11: Additional visualization of different baseline.

Table 6: Influence of baseline choices.

| Baseline | | ImageNet | |
|---|---|---|---|
| $\overrightarrow{x^T x^0}$ | $\overrightarrow{x^0 x^T}$ | Deletion↓ | Insertion↑ |
| **B** | **B** | 0.254 (±0.137) | 0.779 (±0.178) |
| **B** | **W** | 0.272 (±0.137) | 0.743 (±0.166) |
| **B** | **U** | 0.257 (±0.130) | 0.681 (±0.179) |
| **B** | **G** | **0.218 (±0.133)** | **1.130 (±0.168)** |

$\eta$ **setting:** Since smaller $\eta$ needs more computation, we set a balanced $\eta = 0.1$ in Section 4.1. Besides, we set $\eta = 0.02$ for visual comparison because Figure 8 shows that smaller $\eta$ produces more subtle visualizations.

### A.4.3 REIMPLEMENTED BENCHMARK FOR FAIRNESS

Results of different works (Bodria et al., 2021; Petsiuk et al., 2018) are quite distinct. To compare fairly, we reimplement all methods and fix some bugs in the public code. We remove the final softmax layer to avoid interaction of classes and $N = 100$ can already produce stable results. We will also release benchmark code after the paper is accepted.

### A.5 ADDITIONAL EXPERIMENT

### A.5.1 CONSISTENCY ON OUTPUTS AND ATTRIBUTIONS

The ultimate goal of attributions is to monitor the potential mistakes of models and data. We normalize the final scores through a soft-max layer and visualize the attributions of images with high confidence and low confidence shown in Figure 10a and Figure 10b respectively. It can be seen

that for images with high confidence, the corresponding attributions are concentrated on the salient objects, while for images with low confidence, the corresponding attributions are scattered on both the objects and the background.

### A.5.2 ADDITIONAL COMPARISON WITH COUNTERPARTS

To further illustrate the significance of the improvement of the SAMP method compared with other interpretation methods, we show more comparison examples, as shown in Figure 16 and Figure 17. We can observe a clear improvement in the granularity of attributions.

### A.5.3 ADDITIONAL ABLATION STUDY ON BASELINE CHOICES

We show more visualization here to further illustrate that the choice of baselines has a marginal impact in Figure 11. We also test the Deletion/Insertion metrics as Table 6. Different from the visualization results, the choice of baselines will significantly affect metrics. Specifically, for the Deletion/Insertion metrics, we refer to the open-source code[4] from RISE (Petsiuk et al., 2018). In calculating these metrics, RISE uses a black image (i.e., "B") as the baseline for the Deletion metric and a Gaussian blurred image (i.e., "G") as the baseline for the Insertion metric. Therefore, based on the results in Table 6, if we also adopt the "B+G" baseline to generate explanations, which aligns with the metric calculation setup in RISE, the Deletion/Insertion metric would be best.

### A.5.4 ADDITIONAL VISUALIZATION ON DIFFERENT BOUND $\eta$

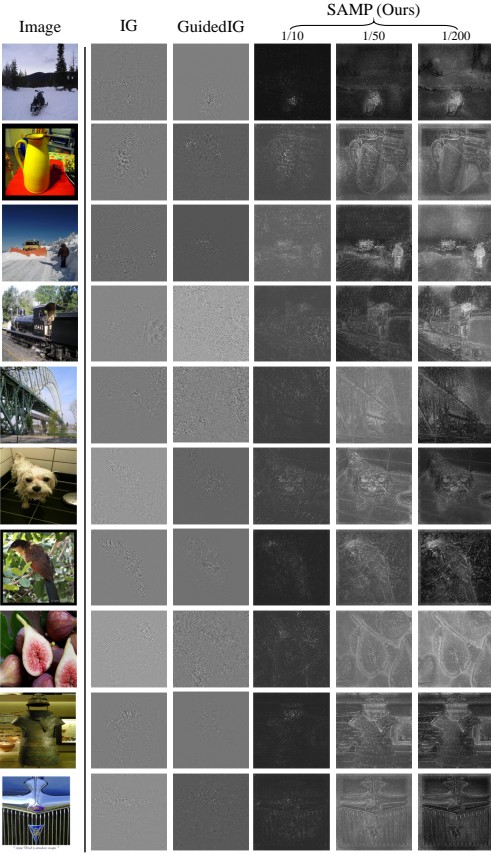

Figure 12: Additional visualization with different upper bound $\eta$.

Here we show more visualization results with different upper bound $\eta$, and compare them with the mainstream IG (Sundararajan et al., 2017) and GuidedIG (Kapishnikov et al., 2021), as shown in

---

[4] https://github.com/eclique/RISE

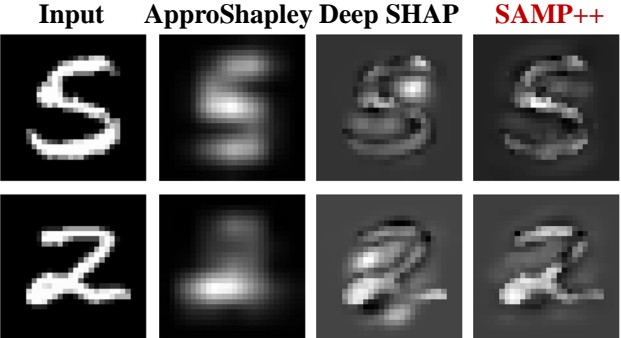

**Input    ApproShapley Deep SHAP    SAMP++**

Figure 13: Visual comparison with Shapley values.

Figure 12. We can see that SAMP can locate salient pixels more clearly, and as $\eta$ decreases, the fineness of localization is higher.

### A.5.5    OTHER EVALUATIONS

**Why Deletion/Insertion:**    Explainability evaluations are still an open problem (Bodria et al., 2021). It is controversial to compare explanations against ground truth since there is no guarantee that models use the same features as humans. Therefore, we choose to employ self-comparison evaluations (e.g., Deletion/Insertion), even though they have potential information leakage.

$\mu$**Fidelity and Pointing game results:**    Despite the controversy, we conduct pointing game experiments for the test set of PASCAL VOC2007 shown in Table 7. We can see that RISE is the most accurate method. RISE has an inherent advantage in pointing game because the block-level methods like RISE (with 7x7 resolution) are not sensitive to outliers. Compared with other path methods (IG, BlurIG and GuidedIG), SAMP still has a significant improvement. We also evaluate the $\mu$Fidelity on MNIST. We can see that SAMP++ achieves competitive performance compared with other methods.

Table 7: $\mu$Fidelity on MNIST and pointing game on VOC2007.

| | MWP | SG | RISE | IG | BlurIG | GuidedIG | SAMP++ |
|---|---|---|---|---|---|---|---|
| $\mu$**Fidelity**↑ | - | 0.327 | 0.286 | **0.453** | 0.292 | **0.453** | **0.461** |
| **Pointing Acc.**↑ | 76.90 | - | **87.33** | 76.31 | 76.19 | 72.82 | **80.75** |
| **Time(s)** (on MNIST) | - | 0.204 | 0.191 | 0.207 | 0.205 | 0.344 | 0.235 |

### A.5.6    COMPARISON WITH SHAPLEY VALUES

Table 8: Comparison with Shapley-based methods on MNIST.

| | ApproShapley | Deep SHAP (Lundberg & Lee, 2017) | SAMP++ |
|---|---|---|---|
| **Deletion**↓ / **Insertion**↑ | 0.014 / 0.808 | 0.029 / 0.955 | **-0.137** / **1.050** |
| **Time(s)** (on MNIST) | 14.390 | 0.748 | 0.235 |

we compare SAMP with two Shapley-based methods. ApproShapley computes the Shapley value by the Monte Carlo method and Deep SHAP (Lundberg & Lee, 2017) is a high-speed approximation. Tab.8 shows that SAMP++ achieves better Deletion/Insertion with less computation burden and Fig.13 visualizes the attributions.

### A.5.7    VISUALIZATION ON MULTI-LABEL CLASSIFICATION

To comprehensively explore the application for more complex scenarios, we constructed 2x2 image combinations using images from the MNIST dataset, where each image contained multiple digits.

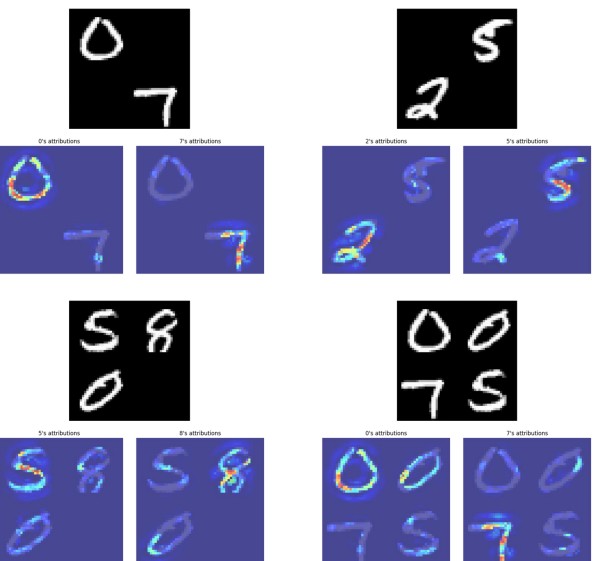

Figure 14: Visualizations on multi-label classification.

Then, we utilized CNNs to classify these images and applied the SAMP method for visualizing the attributions of each classification. The visualization results in Figure 14 demonstrate that SAMP can accurately pinpoint specific data related to certain categories (for example, in synthetic images containing 0, 5, and 7, when attributing to the 0 class, SAMP only attributes to the 0 digit). This suggests that the SAMP method can be effectively applied to multi-class classification problems.

### A.5.8 VISUALIZATION ON FINE-GRAINED DATASET: CUB-200-2011

The CUB-200-2011 (Wah et al., 2011) dataset is a widely used fine-grained bird dataset consisting of 200 types of birds. We employ a ResNet50 network pre-trained on the ImageNet dataset to fine-tune on the CUB-200-2011 dataset until training convergence, achieving an accuracy rate above 98% on the training set. To evaluate the impact of the SAMP method on fine-grained data, we randomly select 5 images from various categories in the CUB-200-2011 dataset and compute attributions on these images with their corresponding 5 category labels, generating a total of 25 visualized images arranged in a 5x5 grid, as shown in Figure 15. In this figure, the **diagonal** is outlined in green borders, representing the attribution result for the image's **corresponding category**, while **non-diagonal** elements are outlined in red borders, denoting attribution results for **other categories**. It is evident that the attribution results for the image's corresponding category are significantly more prominent compared to those for other categories. However, we acknowledge the presence of some intensity in the attribution results of other categories, suggesting that attributions for fine-grained datasets have not yet fully decoupled from the category, which is also a futher direction we aim to further investigate.

### A.6 LIMITATIONS AND FUTURE DIRECTIONS

In this section, we will briefly outline the limitations of the SAMP approach and suggest potential future research directions.

- **Theoretically**, our paper presents the concentration principle to address the issue of ambiguous path selection in traditional path methods. However, directly solving the concentration principle involves an exponentially complex combinatorial optimization problem. To address this, we proposed the SAMP algorithm based on the Brownian motion assumption, which reduces the exponential complexity to linear complexity and can obtain a near-optimal solution to the original problem. Nonetheless, **this paper does not provide a thorough investigation of the optimality of the SAMP algorithm** in solving the

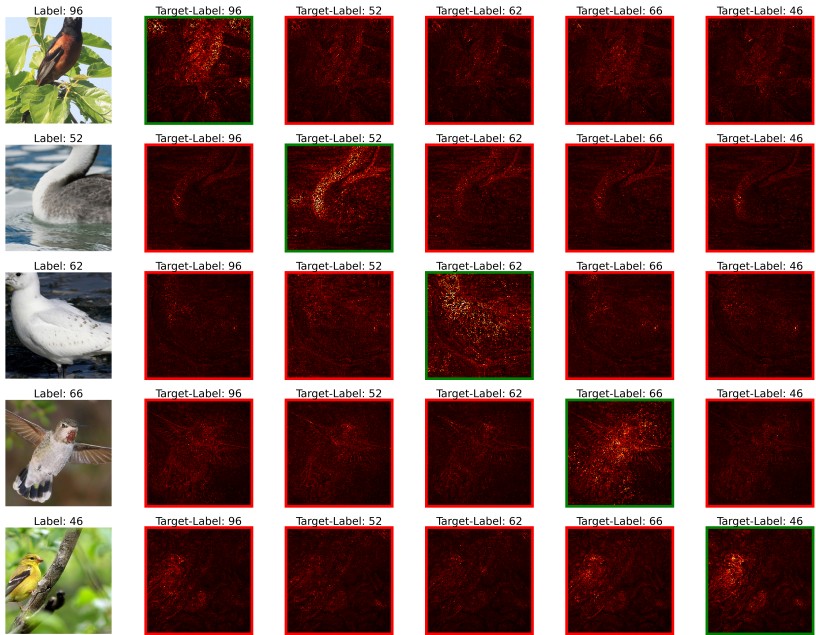

Figure 15: Visualizations on CUB-200-2011.

problem, and we intend to conduct rigorous theoretical analysis of the optimality of the SAMP algorithm in the future and explore if more efficient and accurate algorithms exist for solving the problem.

- **Experimentally**, the SAMP algorithm reduces the original exponentially complex problem to linear complexity and maintains comparability with mainstream path methods (e.g., IG, BlurIG, GuidedIG). However, **the SAMP algorithm still requires multiple forward and backward passes for each sample, leading to a certain computational overhead in practical applications**, and its calculation speed is slower than CAM and Grad-CAM. In the future, we plan to explore if there are more efficient approximation algorithms that can improve execution efficiency while maintaining explanation accuracy.

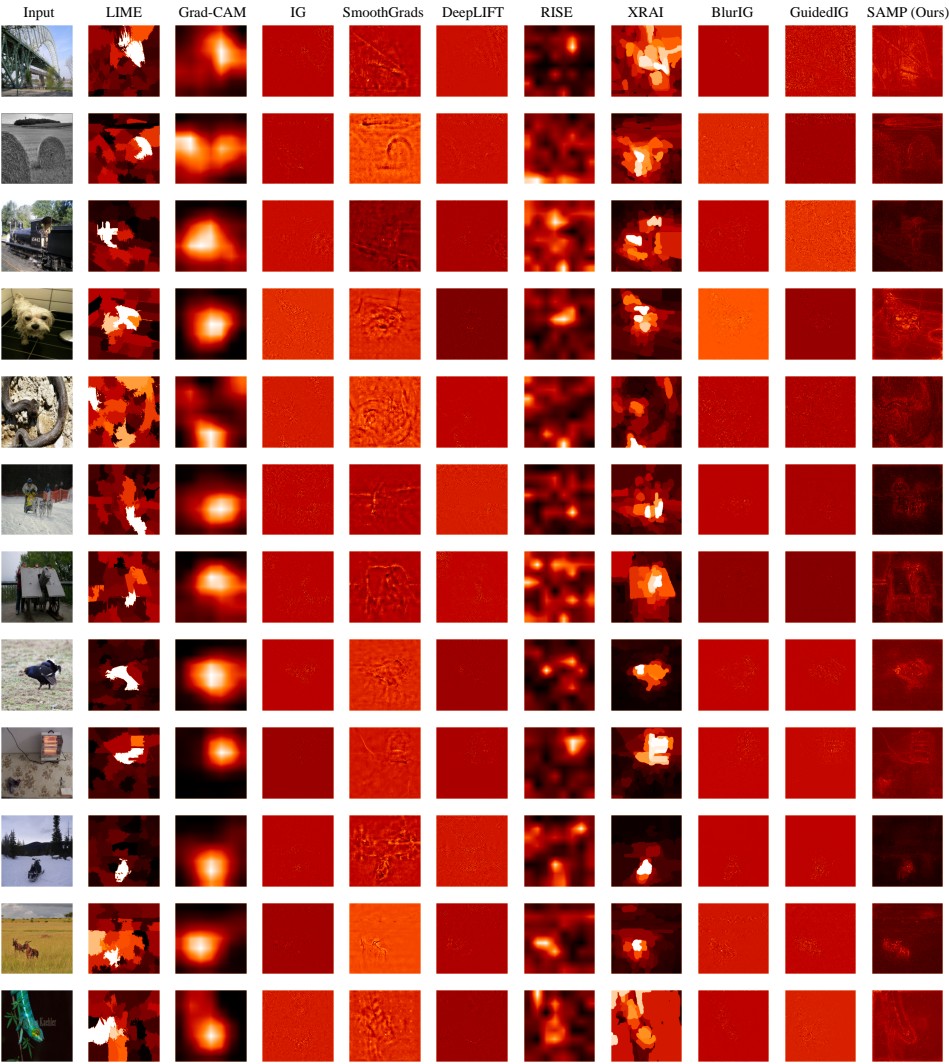

Figure 16: Additional visualization comparison with counterparts.

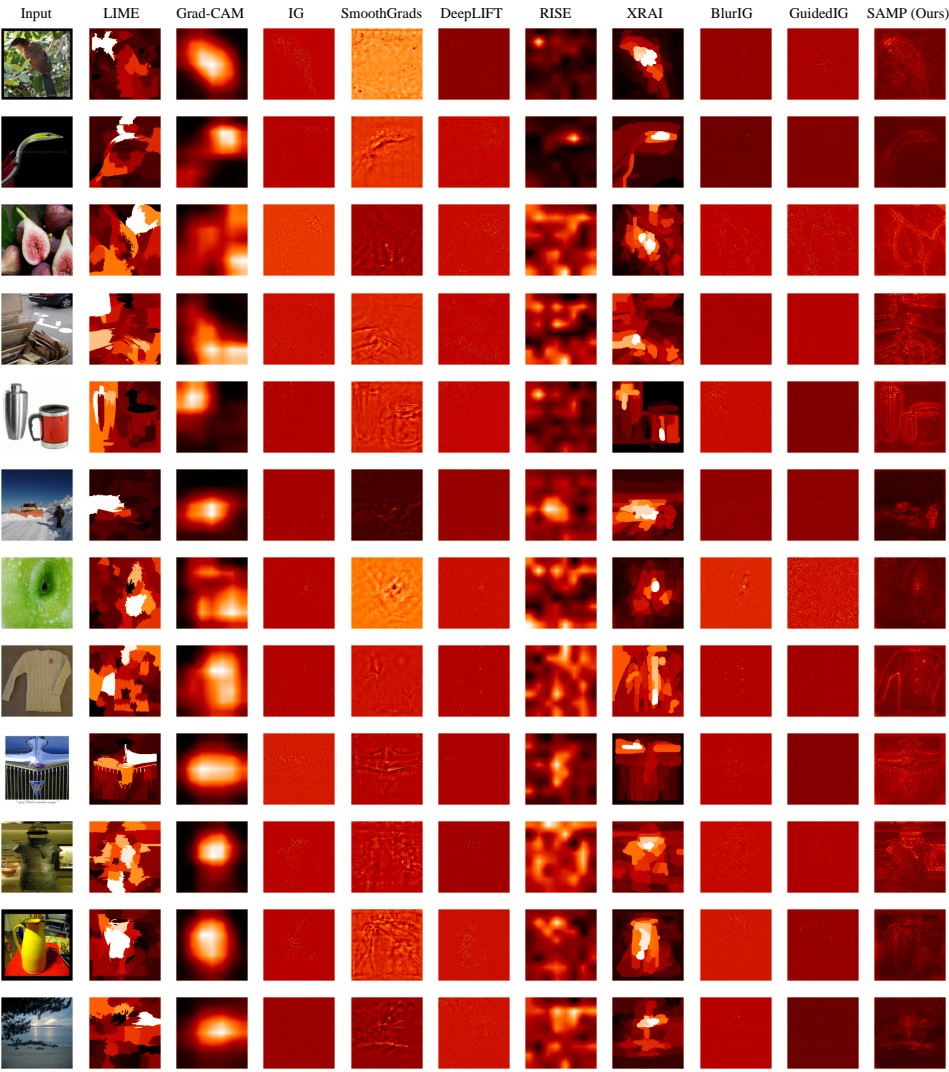

Figure 17: Additional visualization comparison with counterparts.

