# OpenReview forum: "Path Choice Matters for Clear Attributions in Path Methods"
_ICLR.cc/2024/Conference — ICLR 2024 poster_

### Official Review · Reviewer_7nkz · 2023-10-25

**Soundness:** 3 good
**Presentation:** 4 excellent
**Contribution:** 3 good
**Rating:** 8
**Confidence:** 3

**Summary:**

This work studies the efficacy of optimal path to achieve clarity of attributions. A Concentration Principle is defined to guides the interpreter to identify essential features and allocate attributions to them.

**Strengths:**

+ This work study how optimal path can improve the clarity for attributions for interpretations of DNNs. A Concentration Principle is proposed, which aim to concentrate the attributions on the dispensable features.
+ A Salient Manipulation Path approach is proposed to search the near-optimal path (as an approximation of Equation 5). Each manipulation step is constrained below an upper bound with l1-norm, and momentum strategy is proposed to avoid convergence to local solution.
+ This paper shows quantitative and qualitative results to demonstrate the effectiveness of the proposed method.
+ The paper is written clearly and easy to follow.

**Weaknesses:**

- There are some minor typos and latex error for the manuscript.

**Questions:**

- The proposed approach is evaluated on image classification dataset, where there common has a dominant object in the image. For some of the example shown, the attributions could cover several objects (e.g., socks and heaters are both attributed; the dog, tiles line, and shower head are attributed). Does it infer that all the attributed regions/pixels contribute to the decision? Has the author applied this method on fine-grain image classification dataset?
- Brownian motion is the erratic motion of particles suspended in a medium due to continuous collisions. Assumption 1 assume the additive process as the Brownian motion. Please explain why is this valid and the intuition behind.
- The manipulation path is pre-defined. How is the path defined? Could these paths sufficiently cover most scenarios for the search of near-optimal path?
- What is the full name for IG?

Minor comments:
- Some of the figure (or table) is too far from the text that discussed its result, hence not friendly to cross reference the figure and the discussion.

---

> ### Author Response · Authors · 2023-11-23
> **Response to Reviewer 7nkz (Part 1/2)**
>
> We thank reviewer 7nkz for the insightful suggestions on the details of the paper. We have carefully considered each suggestion and made appropriate revisions to the original paper.
> For the revisions of the paper, we have marked them in **green** in the PDF.
>
> ## Weaknesses
>
> **W1:**
>
> Thank you for the feedback from the reviewer. We have double check the content of the paper and have rectified the typos and other mistakes.
>
> ## Questions
>
> **Q1:**
>
> Thanks for the reviewer's suggestion on extensive experiments. In response, we have provided additional information on the visualization results of SAMP in our supplementary materials, specifically addressing the following 3 points:
> - **About attribution across multiple objects:** To address concerns regarding attribution spreading across multiple objects in **Figure 10**, we performed a detailed description in **Appendix A.5.1**. We compared attributions for samples with high and low classification scores (**Figures 10a and 10b**, respectively) and observed that attributions for samples with higher classification scores (i.e., higher model confidence) was more focused, while attributions for lower scored samples (indicating model uncertainty) could potentially spread across multiple objects (such as a dog, tile lines, or shower head). This finding supports the notion that SAMP can reveal uncertainty in the model's data.
> - **Multi-category attribution:** To further explore the potential applications of SAMP, we conducted an experiment detailed in **Appendix A.5.7** involving the splicing of images from the **MNIST** dataset into **2x2 grids**, each containing at least **two digits**. These images were then classified using a CNN before being subjected to attribution via the SAMP method for various categories. Our findings indicate that SAMP can accurately locate only the specified category of data within a given image, even when presented with a mixture of different digit classes. This suggests that SAMP may be **applicable to multi-category** classification problems.
> - **Fine-grained attribution:** Finally, we conducted experiments on the **CUB-200-2011** fine-grained bird classification dataset to further evaluate SAMP's performance on complex data. Details of this experiment can be found in **Appendix A.5.8**. Here, we randomly selected 5 images from the dataset and cross-attributed their respective categories, producing a total of **25 attribution visualizations** displayed in a **5x5 grid** format. Our observations revealed that attribution results for the correct categories were more pronounced, supporting the effectiveness of SAMP for fine-grained data.
>
> **Q2:**
>
> We mainly elaborate on the rationality of Brownian motion assumption in the SAMP method from four aspects.
> - **Intuition behind Assumption 1:** We believe that the allocation of attributions should be a random variable in each step **without any constraints**, and there should be independence among allocations. Without special knowledge about the internal structure of the model, we give the assumption of **isotropy in each allocation**, i.e., each allocation follows an independent and identical Gaussian distribution.
> - **The relationship with Proposition 1:** It is noteworthy that **Proposition 1** has significant differences from **Assumption 1**. Assumption 1 is the **joint distribution** of the isotropic Gaussian assumption under the situation where **allocations have no constraints**, while the real allocation process is **under the condition $\sum_{i=1}^d a_i = C$**, so **Proposition 1** explores the **conditional distribution** among attributions under this condition.
> - **The universality of this assumption:**. Brownian motion (also known as Wiener process) originally refers to the erratic motion of particles suspended in a medium due to continuous collisions, which is a concept in physics. Thereafter, its mathematical model has been widely applied in various fields such as queueing theory, insurance, and computer science. For example, recent popular **diffusion models**[2] make a **Brownian motion assumption** on the forward process for image inputs and achieve excellent image generation performance. In this work, we use the Brownian motion assumption as a tool to **reduce computational complexity** and experimentally verify the effectiveness of this assumption.
> - **The consistency of experimental results:** Based on the Brownian motion assumption, we derived **Proposition 1**, and **Eq(6)** of Proposition 1 demonstrates that $\mathbb{E}(u_k|u_d=C) =  kC / d$, implying that a randomly selected path tends to result in **linear changes** (increase or decrease) in output scores. This phenomenon was also observed in our experiments, as shown in **Figure 1**. We discovered that most other methods (except SAMP) exhibited a trend of **linear change** in their Deletion/Insertion curves, which aligns with our theoretical assumptions and further validates the rationale of this assumption.

---

> ### Author Response · Authors · 2023-11-23
> **Response to Reviewer 7nkz (Part 2/2)**
>
> **Q3:**
>
> Thanks for the suggestions. We have added the following supplementary information related to path concepts:
> - **About Path definition:** Our definition of path primarily follows that in **Integrated Gradients (IG)**[1], as also detailed in **Section 3.1** of our paper. In brief, a path refers to a integral path along which the model output changes. For instance, in the case of deletion direction, the starting point is the original image, and the ending point is the black/noisy image.
> - **Rationality of Manipulation Path:** We mainly explain the rationality from two aspects:
>   - **Relationship between Manipulation Path and Shapley Values:** Shapley values[3] are a classic axiomatic explanation method. Recent work[4] demonstrates how to apply Shapley values to generate explanations in machine learning problems, where a typical scoring function is **removing features with baseline values**, which involves removing partial data features and replacing them with baseline values. This is fundamentally **consistent** with the definition of the Manipulation Path in **Definition 2**.
>   - **Relationship between Manipulation Path and Some Classic Paths:**
>     - The classic path method **IG**[1] chooses a straight line connecting the starting and ending points, which can be seen as a special case of the Manipulation Path with infinitesimal constraints of a very small upper bound $\eta$ and the update step as $s=d$.
>     - Additionally, the **Insertion/Deletion metric** proposed in RISE[5] also utilizes an operation involving replacing partial pixels with baseline values at each step, which is fundamentally **consistent** with the definition of the Manipulation Path.
>
> **Q4:**
>
> We apologize for the confusion about this terminology.
> IG's full name is **Integrated Gradients**[1]. We have clarified the terminology of this method in the **introduction** section of the revised version of the paper.
>
> ## Minor comments
>
> **C1:**
>
> We apologize for the positioning of the figures and tables in the text, which may have caused confusion while reading the paper. We have adjusted the layout in the revised version of the paper.
>
>
> ### Reference:
>
> [1] Sundararajan M, Taly A, Yan Q. Axiomatic attribution for deep networks, ICML 2017
> [2] Ho J, Jain A, Abbeel P. Denoising diffusion probabilistic models, NeurIPS 2020
> [3] Shapley L S. A value for n-person games, 1953
> [4] Chen H, Covert I C, Lundberg S M, et al. Algorithms to estimate Shapley value feature attributions, Nature Machine Intelligence 2023
> [5] Petsiuk V, Das A, Saenko K. Rise: Randomized input sampling for explanation of black-box models, BMVC 2018

---

> ### Author Response · Authors · 2023-11-23
> **Look Forward to further Comments and Suggestions**
>
> Thanks for the extremely valuable suggestions from the reviewer 7nkz. We have provided specific responses to each concern and implemented corresponding adjustments in the original paper. We enthusiastically welcome further remarks and suggestions on our submission and responses.
>
> If the reviewer feel that our responses have sufficiently addressed the concerns, we kindly hope you could keep the rating accordingly. Thank you very much!

---

### Official Review · Reviewer_tbYK · 2023-10-29

**Soundness:** 3 good
**Presentation:** 3 good
**Contribution:** 3 good
**Rating:** 8
**Confidence:** 3

**Summary:**

This paper proposes a new attribution method. For the baseline of integrated gradient, the author proposed the Concentration Principle and also proposed the SAMP method, which can target the black box model. Advantages over partial IG methods on the challenging Imagenet dataset.

**Strengths:**

- A new path attribution principle is proposed for path methods (IG class methods).
- This method can target agnostic models.
- On the challenging imagenet dataset, this method has better attribution effects compared with traditional attribution algorithms and IG-type attribution algorithms.
- This method has theoretical guarantee.

**Weaknesses:**

- Should "Attributions" be changed to "Attribution" in the title of this paper?
- It is not recommended to cite references in the abstract.
- Since the proposed method is attributed to the pixel level, I hope the author can discuss the advantages of this method compared to the method attributed to features, because it seems that the explanation attributed to the region level is more convenient for human understanding.
- In addition to natural image datasets such as Imagenet, I suggest that the author can try to apply this method to natural language interpretation or medical images. Because a single word in natural language has strong semantics and is easy for humans to understand, it is most critical for medical images to have small areas. This can better reflect the practical application value of this article.
- The author mentions the case of model agnosticism. If we consider that the internal gradient of the model is accessible, that is, a white-box model, can the method proposed in this article achieve better attribution results? Or why doesn't it work with white box? I hope the author can discuss relevant content.
- I hope the author can discuss the limitations of this article and future outlook.
- In Table 1, why does the value of deletion appear negative? Did the authors fail to normalize the network's classification output?

**Questions:**

Please see weaknesses, if the author can convincingly address my concerns, I'm open to raise my score.

---

> ### Author Response · Authors · 2023-11-23
> **Response to Reviewer tbYK (Part 1/2)**
>
> We thank reviewer tbYK for the careful review and insightful suggestions.
> We have carefully considered each point and made targeted revisions to the original paper accordingly.
> For the revisions of the paper, we have marked them in **green** in the PDF.
>
> ## Weaknesses
>
> **W1:**
>
> Thanks for the reviewer's suggestion. We have corrected "Attributions" to "Attribution" in the revised version.
>
> **W2:**
>
> Thanks for the suggestion on formatting. We have removed the reference citations in the abstract.
>
> **W3:**
>
> Thanks for the insightful questions from the reviewer. Our responses are as follows:
> - **About the level of Path method explanation:** We agree with the reviewer that feature/region-level explanations are often more understandable. However, we also want to emphasize that the path method is an explanation technique that is **independent of the explanation level**, and can theoretically be applied at both the pixel level and the feature level. For instance, one can change the path $\gamma(\phi)$ in **Eq(1)** from the pixel space to the feature space.
>   - When conducting literature survey, we discovered that some studies (e.g., **XRAI**[4]) have explored how to apply the path method to the feature/region level explanation. Therefore, our study did not focus on the level of explanation, but rather, it primarily investigated the method itself.
> - **About future directions:** Thanks for the reviewer's suggestions. We also believe that explanations at various levels can enhance the diversity of explanations. In our future research plans, we are inspired to integrate existing regional explanation techniques (e.g., XRAI[4]) with the proposed SAMP and provide it to other researchers in the form of an open-source code library.
>
> **W4:**
>
> We find that this is an issue that all three reviewers are concerned about.
> But we have to admit that verifying its effectiveness on NLP within a limited time poses a challenge since the SAMP method was not originally intended for such tasks.
> Moreover, our expertise in medical imaging is somewhat limited, leaving us unsure of the specific requirements for image interpretation in this field. It would be greatly appreciated if the reviewer could recommend relevant literature on this topic.
> We will consider these 2 points in our future work.
> Regarding the reviewer's inquiry regarding the applicability of SAMP in real-world scenarios, we have carried out supplementary testing on two more intricate situations, as detailed below:
> - **Multi-category attribution:** To further explore the potential applications of SAMP, we conducted an experiment detailed in **Appendix A.5.7** involving the splicing of images from the **MNIST** dataset into **2x2 grids**, each containing at least **two digits**. These images were then classified using a CNN before being subjected to attribution via the SAMP method for various categories. Our findings indicate that SAMP can accurately locate only the specified category of data within a given image, even when presented with a mixture of different digit classes. This suggests that SAMP may be **applicable to multi-category** classification problems.
> - **Fine-grained attribution:** We conducted experiments on the **CUB-200-2011** fine-grained bird classification dataset to further evaluate SAMP's performance on complex data. Details of this experiment can be found in **Appendix A.5.8**. Here, we randomly selected 5 images from the dataset and cross-attributed their respective categories, producing a total of **25 attribution visualizations** displayed in a **5x5 grid** format. Our observations revealed that attribution results for the correct categories were more pronounced, supporting the effectiveness of SAMP for fine-grained data.

---

> ### Author Response · Authors · 2023-11-23
> **Response to Reviewer tbYK (Part 2/2)**
>
> **W5:**
>
> Thanks for raising this question about white-box.
> We apologize for not being clear about this in the original paper.
> - **About the nature of the proposed method:** The SAMP method proposed in this paper belongs to **Path Method**, which is related to related works, such as **IG**[1], **BlurIG**[2], and **GuidedIG**[3]. This category of methods **do not** rely on internal variables or gradients of the model, making them distinct from the white-box methods mentioned by the reviewer.
> - **About the advantages of path methods:** There are two key benefits to not considering model gradients.
>   - **Firstly**, for large and complex models, accounting for all internal variables would significantly increase computational overhead.
>   - **Secondly**, focusing solely on the input and output characteristics of the model allows for **implementation invariance** in explanation methods, meaning that for different model implementations, as long as the external input and output characteristics remain the same, the generated explanations will be consistent. This property has been highlighted in the previous research[1] (i.e., Integrated gradients).
> - **About compatibility with white-box methods:** To be honest, as our motivation was **not to consider** variables within the model (as outlined in the "About the advantages of path methods" above), our method does **not natively support** incorporating gradients within the model. However, the reviewer's suggestion has given us food for thought. We can explore taking gradients within the model as an additional input in future work and investigate whether it can enhance the performance of post-hoc interpretable methods.
>
> **W6:**
>
> We apologize for omitting the discussion about limitation in the original paper, and we have added it in **Appendix A.6** of the revised version.
> - **Theoretically**, to address the problem of ambiguous path selection in conventional path methods, this paper proposes the **concentration principle**, yet directly solving the concentration principle is an exponentially complex combinatorial optimization problem. To address this, we proposed the SAMP algorithm based on the Brownian motion assumption, which reduces the exponential complexity to linear complexity and can obtain an near-optimal solution of the original problem. **However**, this paper does **not thoroughly investigate the optimality** of the SAMP algorithm in solving the problem, and we plan to conduct rigorous theoretical analysis of the optimality of the SAMP algorithm in the future and explore if more efficient and accurate algorithms exist for solving the problem.
> - **Experimentally**, although the SAMP algorithm reduces the original exponentially complex problem to linear complexity, ensuring comparability with mainstream path methods (such as IG[1], BlurIG[2], GuidedIG[3]), the SAMP algorithm still requires multiple forward and backward for each sample, so there is still a certain computational overhead in practical applications, and the calculation speed is **slower than CAM and Grad-CAM**. We plan to explore in the future if there are more efficient approximation algorithms that can improve execution efficiency while ensuring the accuracy of the explanation.
>
> **W7:**
>
> Sorry for the confusion.
>
> - **About the curve calculation method:** We have provided details about evaluation methods in **Appendix A.4.1** of the original paper. To guarantee the comparability of curves generated by different methods, it is essential to ensure that the starting and ending points of these curves **remain consistent**, rather than simply normalizing them to 0~1 directly (as the maximum or minimum values of the model output scores might not occur at the start or end). This could result in Deletion curves having values under 0. We encourage the reviewer to read the related sections in the appendix to gain a better understanding of our calculation technique. If the reviewer have any queries about this, we would also be more than happy to engage in further discussions.
> - **On why the maximum/minimum value might not be at the start/end:** We elaborated on this phenomenon in **Section 4.2.1** of the original paper. In brief, on the Deletion curve, as crucial pixels are eliminated progressively, what's left might only be background pixels, leading to a steep decrease in corresponding scores, even below the end point, thus generating scores under 0.
>
> ### Reference:
>
> [1] Sundararajan M, Taly A, Yan Q. Axiomatic attribution for deep networks, ICML 2017
> [2] Xu S, Venugopalan S, Sundararajan M. Attribution in scale and space, CVPR 2020
> [3] Kapishnikov A, Venugopalan S, Avci B, et al. Guided integrated gradients: an adaptive path method for removing noise, CVPR 2021
> [4] Kapishnikov A, Bolukbasi T, Viégas F, et al. Xrai: Better attributions through regions, ICCV 2019

---

> ### Author Response · Authors · 2023-11-23
> **Look Forward to further Comments and Suggestions**
>
> Thanks for the insightful suggestions provided by the reviewer tbYK. We have given targeted responses to each concern and made relevant adjustments to the original paper. We are more than willing to receive further remarks and recommendations from the reviewer on our submission and responses.
>
> If the reviewer feels that our response has adequately addressed each concern, we respectfully request you to raise the rating of our paper. Thank you very much!

---

> > ### Comment · Reviewer_tbYK · 2023-11-23
> > **Official Comment by Reviewer tbYK**
> >
> > Thanks to the authors for preparing a detailed response, I really appreciate it. While the authors have not tested this method on NLP tasks or medical image tasks, the validation conducted on the ImageNet and CUB-200 datasets suggests that this method holds significant value and potential.
> >
> > However, I would recommend that the authors consider adding a sentence on future outlook in their conclusion, specifically addressing the potential application of this method to NLP tasks and medical image tasks. This addition could provide valuable insights into the method's applicability and future research directions. You may also have a look at this paper [1].
> >
> > I have read the comments of other reviewers and I think the author has also addressed most of my concerns. So I decided to increase my score to 8 points.
> >
> > [1] https://www.techrxiv.org/articles/preprint/IG2_Integrated_Gradient_on_Iterative_Gradient_Path_for_eXplainable_AI/24114489

---

> > > ### Author Response · Authors · 2023-11-23
> > > **Response to Reviewer tbYK**
> > >
> > > We express our gratitude for the valuable feedback provided by the reviewer tbYK. In light of these suggestions, we have incorporated plans for future research on natural language processing and medical imaging into the conclusion section of our paper. Your positive appraisal of our work is highly appreciated.

---

### Official Review · Reviewer_4ZSP · 2023-10-31

**Soundness:** 3 good
**Presentation:** 3 good
**Contribution:** 3 good
**Rating:** 6
**Confidence:** 3

**Summary:**

This paper introduces SAMP (Salient Manipulation Path), a novel model-agnostic method for attribution explanation that selects a near-optimal path guided by the proposed Concentration Principle. The authors further enhance SAMP with two additional modules, IC and MS, creating an extended version SAMP++. Performance evaluations of SAMP/SAMP++ are conducted on MNIST, CIFAR-10, and ImageNet datasets using Deletion/Insertion metrics. It outperforms other baseline methods. Additionally, the qualitative results of SAMP and SAMP++ show improvements in visualization quality, offering more precise object location and detailed pixel highlighting. These results demonstrate the advantages of the proposed method.

**Strengths:**

1.	The paper introduces a novel technique for determining the optimal path for distinct attribution allocations, using Brownian motion as the guiding mechanism. To satisfy the completeness axiom, the Infinitesimal Constraint is employed, while the Momentum Strategy is utilized to avoid local optima. Overall, the algorithm is well-motivated and clearly explained.

2.	The evaluation is thorough, and the results highlight the fidelity advantages of both SAMP and SAMP++. Importantly, the efficiency of SAMP and SAMP++ remains relatively high when compared to other integrated gradient methods.

3.	The paper is well-organized and clearly written, contributing to its overall quality.

**Weaknesses:**

1.	Some parts need more clarification:

    a.	Figure 8 states that "as η decreases, the attribution visualization becomes fine-grained." However, at η=10, the visualization appears overly dark, making it challenging to understand. Similarly, Table 3 suggests that “since the main role of IC is to ensure rigorousness, its effect in improving performance is not significant”. Have the authors conducted a statistical test to substantiate this claim?

    b.	In Section 4.4.2, more details should be added, for instance, “B+W” refers to Bin the path x^T to x^0 and W refers to x^0 to x^T. In Figure 7, the authors claim that “the visual impact of different baselines is not significant”. But Figure 7 demonstrates quite different qualitative results. For instance, B+W is in general less salient than others. Moreover, quantitative results in Table 6 in the appendix also show significant differences between baselines. Could the authors elaborate on their reasoning?

    c.	Does SAMP++ provide similar qualitative results as SAMP? Why not present qualitative results of SAMP++ as it has better performance than SAMP?


2.	The current results support the efficacy of the proposed method, but the paper would benefit from more impactful and insightful analyses. For example, it would be informative to investigate whether SAMP++ can help humans understand models, particularly on complex tasks involving fine-grained data.

**Questions:**

How do we understand the impacts of baselines on two paths? To be concrete, in Appendix A.5.3, “the setting of Deletion/Insertion is consistent with ‘B+G’”. How are they related?

---

> ### Author Response · Authors · 2023-11-23
> **Response to Reviewer 4ZSP**
>
> We thank reviewer 4ZSP for the affirmation and pertinent suggestions on our work.
> In response to the issues raised by the reviewer, we have made the following revisions and additions.
> For the revisions of the paper, we have marked them in **green** in the PDF.
>
> ## Weaknesses
>
> **W1:**
>
> Thanks for the insightful suggestions on details. We have carefully considered them and implemented appropriate changes to the paper. The detailed responses are as follows:
> - a.Sorry for the confusion caused by the description.
>   - **Firstly**, $\eta$ does not solely refer to the denominator. "as η decreases, the attribution visualization becomes fine-grained" actually means that the finest granularity is achieved when $\eta = 1/200$, not $\eta = 1/10$. We will clarify this in our revised version.
>   - **Secondly**, in the description "its effect in improving performance is not significant", the "performance" refers to the Deletion/Insertion metrics. We apologize for not being explicit about this. Since Deletion/Insertion[2] is fundamentally an heuristic metric, the strictness of the interpretation method is not directly associated with it. Therefore, we observed this phenomenon experimentally (**Table 3** and the "+IC" row in **Table 2**). Thank Inspired by the insightful feedback from the reviewer, we will continue to delve into the underlying causes in future works.
> - b.Thanks for the suggestion on **Section 4.4.2**, we have now clarified the specific meaning of "B+W" in the revised paper. Regarding **Figure 7**, we apologize for the confusion in our description. We found that the impact of different baselines on the explanations is not significant compared to different methods (as shown in **Figure 4**). However, visually, different baselines have little impact on the **contour information** in the explanation results, but they do significantly affect the **overall intensity** of the explanation results (e.g., brightness), which leads to significant differences in appearance. This point has already been added in the revised version.
> - c.We apologize for the confusion caused by certain terms in the original paper. In **Figure 4**, the SAMP actually refers to the results of **adding IC and MS (i.e., SAMP++)**. We have updated **Figure 4** accordingly. At the same time, we have also included the **original SAMP** algorithm without IC and MS for comparison. As can be seen from the comparison, the visualization results of SAMP++ are **broadly similar** to SAMP, but the results of SAMP++ are more fine-grained (for instance, the subject is more separated from the background).
>
> **W2:**
>
> Thanks for the reviewers' suggestions. We find that this is an issue that all three reviewers are concerned about, so we conducted supplementary experiments under more complex settings. The specific content is as follows:
> - **Multi-category attribution:** To further explore the potential applications of SAMP, we conducted an experiment detailed in **Appendix A.5.7** involving the splicing of images from the **MNIST** dataset into **2x2 grids**, each containing at least **two digits**. These images were then classified using a CNN before being subjected to attribution via the SAMP method for various categories. Our findings indicate that SAMP can accurately locate only the specified category of data within a given image, even when presented with a mixture of different digit classes. This suggests that SAMP may be **applicable to multi-category** classification problems.
> - **Fine-grained attribution:** We conducted experiments on the **CUB-200-2011** fine-grained bird classification dataset to further evaluate SAMP's performance on complex data. Details of this experiment can be found in **Appendix A.5.8**. Here, we randomly selected 5 images from the dataset and cross-attributed their respective categories, producing a total of **25 attribution visualizations** displayed in a **5x5 grid** format. Our observations revealed that attribution results for the correct categories were more pronounced, supporting the effectiveness of SAMP for fine-grained data.
>
>
> ## Questions
>
> **Q1:**
>
> Sorry for the lack of clarity in this section. we have revised the original paper in **Appendix A.5.3**. Specifically, for the Deletion/Insertion metrics, we refer to the **open-source code**[1] from RISE[2]. When calculating this metric, RISE utilizes a **black image** (i.e., **B**) as the baseline for the Deletion metric and a **Gaussian blurred image** (i.e., **G**) as the baseline for the Insertion metric.
> Hence, based on the results in **Table 6**, if we also adopt the **B+G baseline** to generate explanations, which maintains consistency with the metric calculation setup in RISE, the Deletion/Insertion metric would be the best.
>
> ### Reference:
>
> [1] https://github.com/eclique/RISE
> [2] Petsiuk V, Das A, Saenko K. Rise: Randomized input sampling for explanation of black-box models, BMVC 2018

---

> ### Author Response · Authors · 2023-11-23
> **Look Forward to further Comments and Suggestions**
>
> We would like to express our heartfelt gratitude for insightful and valuable feedback from the reviewer 4ZSP. We have carefully considered each of your concerns and have implemented targeted adjustments to the original content. We are more than willing to receive further comments and suggestions from you as we continue to improve our work and responses.
>
> Should you believe our revisions have successfully addressed the issues raised, we kindly request you to adjust the scores accordingly. Your support is greatly appreciated!

---

### Meta-Review · Area_Chair_CFo9 · 2023-12-08

**Metareview:**

This paper presents SAMP (Salient Manipulation Path), a model-agnostic method for attribution explanation, extended to SAMP++ with IC and MS modules. Thorough evaluations on MNIST, CIFAR-10, and ImageNet demonstrate superior performance over baselines. Strengths include a well-motivated algorithm, clear presentation, and high efficiency. However, some sections need clarification, and additional impactful analyses, such as human model understanding and diverse dataset applications, would enhance the paper. While strengths include theoretical guarantees and improved attribution effects on challenging datasets, addressing weaknesses like minor errors and discussing limitations and future outlooks would further strengthen the paper.

All reviewers unanimously recommend accept.

Overall, it is a promising contribution to the ICLR community.

**Justification For Why Not Higher Score:**

Though this paper presents an interesting method for DNN explainability, it is merely for conventional architectures but not the most recent large models. So, its usability is yet to be fully confirmed.

**Justification For Why Not Lower Score:**

Good presentation, good paper, all reviewers recommend accept.

---

### Decision · Program_Chairs · 2024-01-16

Accept (poster)